# ENHANCING MULTI-MODAL REASONING OVER TIME-SERIES AND NATURAL LANGUAGE DATA

## ABSTRACT

Time-series analysis is critical in many industries such as healthcare, finance, transportation, and energy sectors. The practical application of time-series analysis often involves analyzing time-series data alongside contextual information in the form of natural language for informed decision making. However, current time-series models are limited in their ability to perform reasoning that involves both time-series data and textual information. In this work we address this gap by introducing *Chat-TS*, a large language model (LLM) designed specifically for reasoning over time-series and textual data. Unlike traditional time-series models Chat-TS integrates time-series tokens into the LLM vocabulary, enhancing its reasoning ability over both text and time-series modalities without compromising its core natural language capabilities, enabling practical analysis and reasoning across time-series data. To support the development and validation of Chat-TS we contribute three new datasets: the *TS Instruct Training Dataset* which pairs diverse time-series data with relevant text instructions and responses for instruction tuning, the *TS Instruct question and answer (QA) benchmark*, a set of nearly 4000 multiple-choice questions designed to evaluate multi-modal reasoning and the *TS Instruct Qualitative Benchmark* which provides a smaller subset of QA, math and decision making questions for LLM evaluation. We have designed a training strategy that preserves the inherent reasoning capabilities of the LLM while augmenting it with time-series reasoning capabilities. Evaluation results show that Chat-TS achieves state-of-the-art performance in multi-modal reasoning tasks, maintaining strong natural language proficiency while advancing time-series reasoning. All models, datasets, and code will be made publicly available [Github URL].

## 1 INTRODUCTION

Time-series data is a valuable form of information, widely used in sectors such as healthcare (Morid et al., 2023), business (Zhang et al., 2024a), and industry (Kashpruk et al., 2023). Time-series analysis, particularly for forecasting, imputation, anomaly detection, and classification, has been extensively studied over several decades (Makridakis et al., 1977; Dempster et al., 2020; Jiang et al., 2024). These pure time-series analysis tasks often make predictions and draw conclusions from time-series data alone. However, the practical application of time-series data often necessitates drawing conclusions and making decisions based on real-world knowledge alongside the time-series data itself. For example, an investment banker may consider both stock prices and relevant company reports, while a doctor may analyze a patient's ECG readings in combination with their medical history and clinical notes.

There has been extensive research into the application of large language models (LLM's), for example as financial trading agents (Wang et al., 2023; 2024a; Li et al., 2023) and medical analysts (Nazi & Peng, 2024; Thirunavukarasu et al., 2023). As part of their analysis these models process domain specific time-series. In these cases the time-series is converted from numerical representations to text format to be processed by the LLMs. Despite the promising applications of LLMs to handle real world multi-domain challenges, the training and evaluation of LLM reasoning abilities, specifically for time-series data, remains significantly under-explored.

In other domains such as computer vision, multi-modal reasoning via LLMs is a quickly growing research area with important works such as LLaVA (Liu et al., 2023), MM1 (McKinzie et al., 2024), Open-Flamingo (Awadalla et al., 2023) among others (Kondratyuk et al., 2024; Zhang et al., 2024b) enhancing LLMs for multi-modal reasoning. Multi-modal capacity has even been adopted by many of the most popular commercial models such as GPT4o (OpenAI et al., 2024), and Gemini (Team et al., 2024a).

Unlike the multi-modal models under development in other domains, current time-series analysis methods are typically limited in their ability to integrate other modalities, such as text, for holistic decision-making. This gap highlights the need for models that can reason across both time-series data and related textual information. In this work, we propose a multi-modal training process which integrates time-series reasoning capabilities into existing LLMs. To demonstrate its effectiveness, we also compare the results to the current state-of-the-art text encoding method introduced by Gruver et al. (2024).

Developing an LLM with time-series(TS) reasoning capabilities presents several key challenges: (1) **Data Scarcity:** There is a lack of existing training data for constructing models that combine time-series data with reasoning. Multi-modal datasets that integrate both text and time-series are currently limited. (2) **Benchmarking:** Similarly, there are no large-scale benchmarks available to evaluate multi-modal reasoning and decision-making for time-series data. (3) **Preserving LLM Capabilities:** Current time-series approaches that rely on LLMs as sources of pre-trained parameters do not preserve natural language knowledge and reasoning capabilities inherent in LLMs. Many methods re-train the input and output layers to directly generate or classify time-series data, which often strip away these important reasoning skills.

To address these challenges, our paper makes the following critical contributions:

- We propose a novel approach to leveraging multi-modal LLMs for time-series analysis, with a specific emphasis on maintaining the LLM's reasoning capabilities and using its vast knowledge reservoir to enhance time-series data reasoning.

- We contribute three different datasets for training and evaluation of LLMs for time-series reasoning tasks. The **TS Instruct Training Dataset**, a diverse multi-modal dataset that integrates time-series data with relevant textual information, addressing the critical scarcity of such datasets. The **TS Instruct QA Benchmark**, a dataset of nearly 4000 multiple-choice questions with ground truth answers, designed to evaluate multi-modal reasoning capabilities in time-series analysis. The **TS Instruct Quantitative Evaluation Benchmark**, a quantitative benchmark of approximately 200 samples for assessing more nuanced reasoning, including QA, math, and decision-making tasks, to provide a more comprehensive evaluation of model performance beyond simple accuracy metrics.

- Finally, we design a new family of **Chat-TS Models** that explicitly preserve the general knowledge and reasoning abilities of LLMs within $\pm 2\%$ of their original capacity on challenging LLM benchmarks, and improve reasoning capabilities by on average $\sim 15\%$ on our new TS Instruct QA benchmark compared to existing methods.

To ensure replicability and facilitate future research all datasets, models, and code for dataset generation, pre-processing, training, and testing are made publicly available at [link].

## 2 RELATED WORK

**Large Language Models in Time-Series**  The analysis of time-series is a well studied field, increasingly leveraging deep learning techniques for tasks such as forecasting, imputation, anomaly detection, and classification (Wu et al., 2023; Liu et al., 2024; Nie et al., 2023). Recently, large language models have been explored for these tasks, but many approaches use them primarily as a source of pretrained parameters without fully utilizing their reasoning capabilities (Zhou et al., 2023; Chang et al., 2024). Some models, such as Time-LLM (Jin et al., 2024), TEMPO (Cao et al., 2024), and TEST (Sun et al., 2024), attempt to integrate text and prompting with time-series data to harness LLMs for time-series analysis. However, these works focus primarily on forecasting and modify the model architecture, removing the text generation capabilities, and preventing their use as general-purpose LLMs for time-series reasoning and analysis.

| Model Type | LLM Core as Model | Text as Input | Text Generation | NLP Capabilities | TS Reasoning |
|---|---|---|---|---|---|
| GPT4TS | ✓ | ✗ | ✗ | ✗ | ✗ |
| LLM4TS | ✓ | ✗ | ✗ | ✗ | ✗ |
| TEMPO | ✓ | ✗ | ✗ | ✗ | ✗ |
| TEST | ✓ | ✓ | ✗ | ✗ | ✗ |
| Time-LLM | ✓ | ✓ | ✗ | ✗ | ✗ |
| InstructTime | ✓ | ✓ | ✓ | ✗ | ✗ |
| LLMTime | ✓ | ✓ | ✓ | ✓ | ✗ |
| **Chat-TS (Ours)** | ✓ | ✓ | ✓ | ✓ | ✓ |

Table 1: Capability comparison of popular pretrained models for time-series analysis with an LLM core. None of these works evaluate their models for time-series reasoning with most (except LLM-Time) losing their ability to reason entirely.

A few recent models have managed to preserve the full capabilities of LLMs for time-series reasoning. For instance, LLMTime (Gruver et al., 2024) encodes time-series data as text, allowing the LLM to perform forecasting without any task-specific fine-tuning. Although their experiments focus primarily on forecasting, the approach preserves the LLM's ability to reason over time-series data, as it does not interfere with the text-based structure of the model. This encoding strategy serves as our state-of-the-art baseline. Another example is InstructTime (Cheng et al., 2024), which fine-tunes LLMs for time-series classification by predicting text labels. While effective for classification, this approach is less suitable for general time-series reasoning due to the task-specific nature of the fine-tuning process.

A more recent work by Merrill et al. (2024) evaluates LLMs specifically for time-series reasoning, using LLMTime text encoding as a baseline, similar to our approach. This study employs synthetic time-series data generated by AI, testing various model configurations. While it underscores the importance of adapting LLMs for time-series reasoning, it does not provide methodologies or training datasets to improve model performance. In contrast, our work addresses these limitations by introducing a comprehensive training strategy and novel datasets that enable LLMs to reason effectively across time-series data without compromising their general language capabilities.

Table 1 summarizes the capabilities of existing models that integrate LLMs with time-series analysis. Most existing approaches, such as GPT4TS, LLM4TS, and TEMPO, use LLMs as their core but fail to support comprehensive multi-modal reasoning, including text input, text generation, and time-series reasoning. For example, models like InstructTime and LLMTime support text input and generation, but they do not explore the models time-series reasoning capabilities. Our proposed Chat-TS model is the first to combine all these elements, enabling both robust NLP and time-series reasoning, thus filling a critical gap in the current literature.

**Time-Series Evaluation** There are many large-scale, diverse and publicaly available datasets for training and evaluation models on time-series tasks, such as Monash Time-Series Forecasting Repository (Godahewa et al., 2021), the Time-Series Classification Archive (Middlehurst et al., 2024) and the LOTSA (Woo et al., 2024) dataset compilation. However, these datasets primarily contain pure time-series data with limited accompanying textual descriptions, making them unsuitable for training multi-modal LLMs that require both time-series and text data for reasoning. To bridge this gap, we augment the LOTSA dataset by adding detailed text descriptions for each time-series. This data can then be used in conjunction with AI to generate high quality synthetic datasets for training multi-modal LLM's for time-series reasoning.

**Discrete Tokenization** Previous work in time-series (Cheng et al., 2024), computer vision (Yu et al., 2024), and audio domains (Défossez et al., 2022; Borsos et al., 2023) typically rely on vector quantization methods like VQ-VAE (van den Oord et al., 2018), which control the compression factor of the input data for performing multi-modality tokenization. While these methods work well for high-dimensional data, most time-series data in the literature have sequence lengths below 1000 points (Zhou et al., 2021; Max-Planck-Institut fuer Biogeochemie; Makridakis et al., 2020; Trindade, 2015; California Department of Transportation, 2024), making reconstruction accuracy

and scalability more important than compression. Even for large multi-dimensional time-series many works use channel independent processes such as patching (Nie et al., 2023) to process each channel independently greatly reducing the total effective input size. For this reason we designed our own simple discrete tokenizer which does not require any training and does not compress the time-series data.

# 3 A FRAMEWORK FOR REASONING OVER TIME-SERIES AND LANGUAGE DATA

## 3.1 PROBLEM DEFINITION

Multi-modal time-series reasoning involves a model's ability to make informed decisions and answer relevant questions based upon time-series data and other non-numerical data such as text.

The model design objective is to expand the vocabulary of a pre-trained LLM to include time-series tokens, training autoregressively alongside text. Let $\mathcal{V}_L$ denote the original textual vocabulary and $\mathcal{V}_T$ the set of time-series tokens, forming an extended vocabulary $\mathcal{V} = \mathcal{V}_L \cup \mathcal{V}_T$. The LLM, parameterized by $\theta$, is trained to predict the next token in a sequence $\mathbf{y} = \{y_1, y_2, \ldots, y_{T-1}\}$, where each $y_t \in \mathcal{V}$, using the autoregressive model $f_\theta : \mathcal{V}^* \to \mathcal{V}$. The training process maximizes the likelihood of the next token via standard cross-entropy (CE) loss: $\mathcal{L}(\theta) = -\sum_{t=1}^{T-1} \log p_\theta(y_{t+1}|y_1, y_2, \ldots, y_t)$, where $p_\theta$ is the predicted probability of the next token. Since $\mathcal{V}$ contains joint representations for text and time-series the training process is reduced to standard LLM training.

The central challenge in developing a high-quality instruction model capable of reasoning with time-series data is balancing the specialized time-series reasoning while maintaining the general natural language understanding (NLU) capabilities of large language models. Our datasets include both Etiological and Question and Answering based time-series reasoning as defined in Merrill et al. (2024). Many existing time-series approaches, such as Zhou et al. (2023), Cao et al. (2024), and Chang et al. (2024), tend to narrow the model's focus by introducing task-specific heads, which are trained over the natural language head and prevent the model from performing multi-modal time-series reasoning tasks. In contrast, our approach seamlessly integrates TS data into the LLM's vocabulary, preserving its powerful NLU capabilities without sacrificing TS reasoning.

## 3.2 OVERVIEW

Training a joint text and time-series model consists of training data, tokenization, and an LLM as core required components. The overall components are shown in figure 1.

Data is the cornerstone of any multi-modal learning approach, and high-quality text and time-series data are critical for training robust models. However, while there are many high-quality datasets available for either text or time-series, there are no existing datasets that integrate both modalities—a significant barrier for instruction-tuning LLMs for time-series tasks. To address this, we contribute a novel dataset designed specifically for this purpose, using a combination of time-series properties, task descriptors, and synthetic conversations generated by GPT4o-mini.

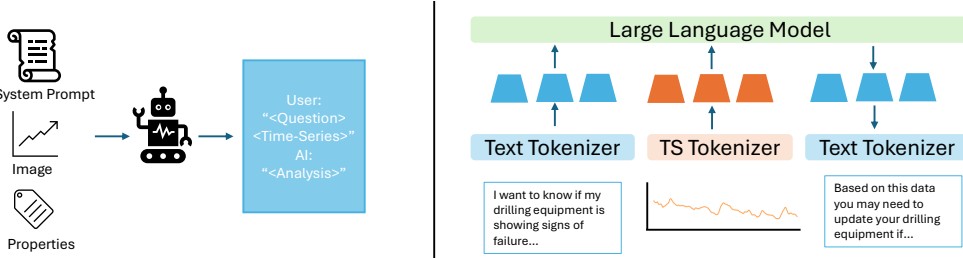

Figure 1: Overview of the TS Instruct Dataset construction followed by the Chat-TS models for time-series reasoning.

We train our model using both text and time-series data, aiming to allow the model to reason over time-series data while preserving its original language modeling capabilities. Unlike other multi-modal models, which require connectors to modify representations for LLM integration (McKinzie et al., 2024), our approach directly adds time-series tokens to the LLM's vocabulary. This eliminates the need for intermediary connecting components between the time-series and text representation spaces, ensuring a more seamless integration. We then explore two initialization methods for the new embeddings: (1) mean initialization, where embeddings are initialized as the average of existing text tokens, and (2) pre-training, where we unfreeze only the embedding and final linear layer for time-series data.

We fine-tune our model with instruction and answer pairs using the multi-modal TS-Instruct dataset alongside a subset of the Open-Orca (Lian et al., 2023) text-only instruction tuning dataset. This dual-dataset strategy helps preserve the LLM's strong language modeling capabilities while also augmenting its ability to understand and generate time-series data. The balanced training approach ensures that the model remains proficient at text tasks while gaining expertise in time-series reasoning.

## 3.3 DISCRETE TOKENIZERS

Tokenizing time-series data presents unique challenges, as it involves converting continuous numerical values into discrete tokens that can be processed by an LLM. Our approach balances reconstruction accuracy with token scalability, ensuring that the tokenized time-series can be efficiently represented without compromising data integrity.

There are several key considerations when designing a time-series tokenizer:

1. **Reconstruction accuracy**: The tokenizer must encode and decode time-series data with minimal error.

2. **Compression**: It should allow for efficient compression to process larger time-series sequences.

3. **Scalability**: The tokenizer should support varying vocabulary sizes, improving quality as the number of tokens increases.

Our discrete tokenizer functions as follows: Let $X \in \mathbb{R}^{L \times M}$ represent a multivariate time-series, where $L$ is the sequence length and $M$ is the number of channels. We quantize the time-series by dividing the range $[-s, s]$ into $K - 1$ bins, mapping each value $x$ to its corresponding bin index $i$, which forms a sequence of discrete tokens $Z$. A special token $T_c = K$ is added to mark the end of each channel, and the token sequence $T$ is then flattened and offset to align with the vocabulary size of the text tokenizer. Decoding reverses this process, reconstructing the time-series by mapping each index back to the bin centers. The special token is used as a channel separator to reconstruct the original 2D time-series from the 1D sequence of tokens.

We tested four tokenizers, including models using simple linear layers for encoding/decoding with vector quantization (VQ) (van den Oord et al., 2018) and lookup-free quantization (LFQ) (Yu et al., 2024), and a TCN-based (van den Oord et al., 2016) tokenizer using residual vector quantization (RVQ) (Chen et al., 2010) as implemented in InstructTime (Cheng et al., 2024). We compare these approaches to our simple discrete tokenizer that requires no training and provides near-perfect reconstruction of time-series data.

**Evaluation** To evaluate the performance of each tokenizer, we measured reconstruction quality using varying compression ratios and codebook sizes on the Azure VM Traces 2017 dataset (Hadary et al., 2020). Figure 2 presents the results, showing that our discrete tokenizer significantly outperforms the others in terms of validation error and improves predictably as the codebook size increases.

The results clearly show that the discrete tokenizer far outperforms the other tokenizers in terms of validation error and predictably improves when scaling the number of codebook tokens. The main downside is that the discrete tokenizer does not compress the time-series. While this is a technical limitation, the input and output capabilities of LLM's is rapidly growing. We therefore feel this is a reasonable limitation to work with. Importantly our tokenizer doesn't require any training and therefore does not suffer from out of distribution problems that may arise from trained tokenizers.

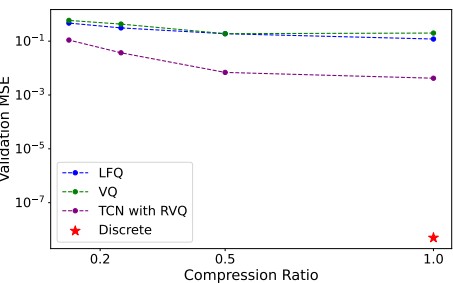
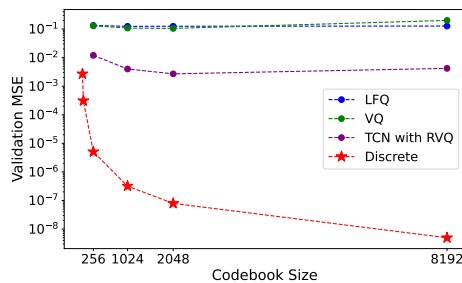

(a) Tokenizer compression ratio vs reconstruction error.

(b) Tokenizer codebook size vs reconstruction error.

Figure 2: Tokenizer validation study. By trading off time-series compression, we are able to build a tokenizer with minimal reconstruction error and predictable accuracy gains with increasing codebook size.

## 3.4 DATASET CREATION

Our training data is categorized into three types: time-series data for pretraining, natural language data, and a new joint time-series and natural language dataset. Full details regarding each data type can be found in Appendix B.

**Time-Series Data:** We leverage the LOTSA (Woo et al., 2024) dataset archive, which spans diverse domains and ensures variation in sequence length and channel dimensions. Time-series samples are generated using a sliding window approach with configurable window size, stride, and channel selection, optimizing for diversity and maximum sample size.

**Natural Language Data:** To maintain and augment performance on natural language tasks, we use 100,000 samples from the Open-Orca (Lian et al., 2023) dataset, which provides diverse high quality natural language instruction tuning data.

**Joint Time-Series and Natural Language Data:** As there is limited multi-modality time-series data available, we synthesize conversations between a user and an assistant using GPT4o-mini (OpenAI et al., 2024). These dialogues incorporate time-series data represented as images alongside metadata (length, channels) to guide the model. Conversations are designed to focus on general reasoning, classification, decision-making, and mathematical analysis. Time-series data for reasoning tasks are sourced from LOTSA (Woo et al., 2024), while classification labels come from the Time-Series Classification Archive (Middlehurst et al., 2024).

## 3.5 TRAINING

We test two methods for integrating time-series tokens: mean initialization, and pre-training with time-series data. Mean initialization sets new time-series embeddings based on the average of text tokens in the model vocabulary, while pre-training unfreezes a small portion of the model (embedding layer and final linear layer) to allow these new embeddings to specialize. Freezing most of the model preserves the LLM's existing capabilities while allowing the new embeddings to learn from time-series data. Next, each model is trained either on our TS Instruct dataset and/or the Open Orca text dataset.

In our experiments we use the 8 billion parameter LLama 3.1 model (Dubey et al., 2024) which provides a reasonable balance between performance and compute. Our pretraining stage consists of 115,000 time-series samples for 1 epoch using an initial learning rate of $2e - 3$. We then instruction tune the model on our dataset comprising roughly 18,000 multi-modal samples and 100,000 text samples with a learning rate of $2e - 5$ for 1 epoch. In both cases we use cosine-scheduling for the learning rate. For more information refer to Appendix C.

## 4 RESULTS AND ANALYSIS

We evaluate our model using the **TS-Instruct QA Benchmark**, a dataset designed to address the lack of benchmarks for multi-modal models capable of time-series reasoning. The benchmark evaluates models through multiple-choice question-and-answer (QA) tasks, where each question includes a description of the time-series data, four answer options, and a detailed explanation of the correct answer. This allows us to assess both answer accuracy and reasoning capabilities. The dataset was generated using GPT-4o-mini, which, combined with visual representations and time-series metadata is used to generate the text components of the multiple-choice questions. The full system prompt is detailed in Appendix D.

To establish a controlled baseline for model evaluation and evaluate the quality of the generated testing data, we curated a high-quality subset of the benchmark, the **TS-Instruct QA Gold dataset**. This dataset consists of 100 random samples from the **TS-Instruct QA Benchmark**. We evaluated the accuracy of these samples for their correctness and found that 92% of the samples were of high quality. These samples were then reviewed and rewritten to ensure clarity, diversity, and accuracy. The Gold dataset serves as a reliable benchmark for time-series reasoning under optimal conditions, providing a fair basis for comparing models.

The evaluation methodology comprises four components: (1) **Baseline Comparison** on the Gold dataset to benchmark reasoning capabilities under controlled conditions; (2) **Natural Language Performance**, evaluating the impact of time-series instruction tuning on the model's general language understanding and reasoning abilities; (3) evaluation on the **Full TS-Instruct QA Benchmark** (3741 samples) to assess the model architecture and training data composition; and (4) **Quantitative Analysis of Generated Explanations**, assessing the quality of responses using metrics such as helpfulness, relevance, accuracy, and level of detail.

### 4.1 BASELINE COMPARISON ON TS-INSTRUCT QA GOLD

We compare our model against baselines using the curated **TS-Instruct QA Gold** dataset. Table 2 presents the results, ranking models by TS reasoning performance and including MMLU-Pro scores to assess general NLP capabilities.

Table 2: Comparison of TS Reasoning and NLP (MMLU-Pro) performance. Models are ranked by TS Reasoning performance. Model baselines are as follows: Gemma-2 2B (Team et al., 2024b), Gemma-2 9B (Team et al., 2024b), Llama 3.1-8B (Dubey et al., 2024), Ministral-8B (Jiang et al., 2023), Phi-3-medium-4k (Abdin et al., 2024), GPT-4o-mini and GPT-4o (OpenAI et al., 2024).

| Model | TS Performance (Correct Avg %) | MMLU-Pro Score | # Parameters (Billions) |
|---|---|---|---|
| Gemma-2 2B | 35.8 | 0.2719 | 2 |
| Gemma-2 9B | 43.4 | 0.4125 | 9 |
| LLama 3.1-8B | 47.4 | 0.3772 | 8 |
| Ministral-8B | 49.6 | 0.3483 | 8 |
| Phi-3-medium-4k | 53.6 | 0.474 | 14 |
| **PreOrcaTS (ours)** | 54.4 | 0.356 | 8 |
| GPT-4o-mini | 66.4 | 0.631 | unknown |
| GPT-4o | 68.2 | 0.766 | unknown |

**Analysis:** This evaluation shows not only the effects of model size but also of different architectures. It also highlights the importance of work such as ours that focus on building models for time-series reasoning analysis. Despite the size of our model, it outperforms model's such as Phi-3-medium-4k-instruct which are both larger and show stronger reasoning capabilities on NLP benchmarks. These results also show the importance of scale since scaling up model performance (for example gemma-2b to gemma-9b) generally increases performance on both the MMLU-Pro and time-series benchmarks. We include in our analysis "closed-source" models such as GPT-4o and GPT-4o-mini. These results should be interpreted with caution as the GPT-4o-mini (which likely shares training data with GPT-4o) was used in the original dataset generation process.

We additionally tested some common visual models for time-series analysis. We have included these results in Appendix E.1. Due to the fact that our dataset comprises real-world time-series and

not synthetically generated time-series, visual representations are used during the dataset generation process to help align the text with the time-series. This causes visual baselines to have inflated accuracy on our benchmark. Recent work in (Merrill et al., 2024) evaluated vision baselines on synthetic time-series data and noted that the visual baselines did not perform better than the models which use text encoding based strategies (Gruver et al., 2024). This is a necessary trade-off in building high quality time-series datasets that use real-world time-series.

## 4.2 Model Validation

We analyze multiple variants of our model to assess the contribution of each training component. This includes studying the impact of using only text data from the Open Orca dataset, the effect of pretraining the added time-series tokens using the time-series dataset described in B.1, and the role of instruction-tuning with the TS Instruct dataset. The abbreviations in Table 3 correspond to each model variant and its training data.

Table 3: Model Variants and Training Components

| Model Variant | Training Components |
|---|---|
| **Orca** | LLama 3.1-8B + Open Orca (Text only) |
| **OrcaTS** | LLama 3.1-8B + Open Orca + TS Instruct |
| **TS** | LLama 3.1-8B + TS Instruct |
| **PreTS** | LLama 3.1-8B + TS Pre-Training + TS Instruct |
| **PreOrcaTS** | LLama 3.1-8B + Open Orca + TS Pre-Training + TS Instruct |

### 4.2.1 Preserving NLP Capabilities in Multi-Modal Models

Previous works (Zhou et al., 2023; Chang et al., 2024) have integrated LLMs as core components of time-series models. However, a common issue with these models is that they often fail to retain the general natural language understanding and reasoning capabilities of the underlying LLM. In contrast, we have preserved the quality of the base LLM through our discrete tokenization strategy and careful training paradigms.

Table 4 presents the performance of different model variants across three natural language benchmarks: MMLU-Pro (Wang et al., 2024b), Big Bench Hard (Suzgun et al., 2022), and GPQA (Rein et al., 2023). The results demonstrate that our multi-modal LLMs maintain comparable performance to the base LLama 3.1-8B model, indicating that integrating time-series reasoning does not degrade the model's natural language understanding capabilities.

Table 4: Comparison of model accuracy for LLAMA 3.1-8B Variants on MMLU-Pro, Big Bench Hard, and GPQA benchmarks. All models are within 2% of each other on each benchmark.

| Training Data | MMLU-Pro | Big Bench Hard | GPQA |
|---|---|---|---|
| LLama 3.1-8B | $0.376 \pm 0.004$ | $0.511 \pm 0.154$ | $0.324 \pm 0.029$ |
| Orca | $0.367 \pm 0.004$ | $0.529 \pm 0.156$ | $0.328 \pm 0.035$ |
| OrcaTS | $0.356 \pm 0.004$ | $0.522 \pm 0.158$ | $0.341 \pm 0.042$ |
| TS | $0.360 \pm 0.004$ | $0.530 \pm 0.161$ | $0.322 \pm 0.032$ |
| PreTS | $0.363 \pm 0.004$ | $0.526 \pm 0.158$ | $0.318 \pm 0.014$ |
| PreOrcaTS | $0.356 \pm 0.004$ | $0.525 \pm 0.155$ | $0.340 \pm 0.042$ |

**Analysis:** Interestingly, even models trained solely on the TS Instruct dataset (i.e., without Open Orca data) achieve performance nearly identical to models trained with interleaved text and time-series data. This stands in contrast to observations in other multi-modal domains, such as computer vision, where models often rely on trained connectors to link visual and textual representations (McKinzie et al., 2024). In such domains, text data remains imperative for preserving the underlying LLM capabilities. Each model is within $\pm 2\%$ of the original base LLama 3.1-8B model. These

differences can be attributed differences in alignment from the full parameter fine-tuning and does no necessarily indicate a increase or decrease in overall model quality.

Our results demonstrate that expanding the model's vocabulary with discrete tokenizers is an effective way to extend LLM capabilities without compromising natural language reasoning performance. This ensures that the model retains its foundational NLP strengths while acquiring the ability to reason effectively with time-series data.

### 4.2.2 TS-INSTRUCT QA BENCHMARK

To evaluate performance at scale, we use the complete TS-Instruct QA dataset containing 3741 samples. This larger evaluation allows us to assess robustness and instruction-following capabilities across varied prompt configurations. The evaluation was repeated 5 times with different system prompts which can be found in Appendix D.3.

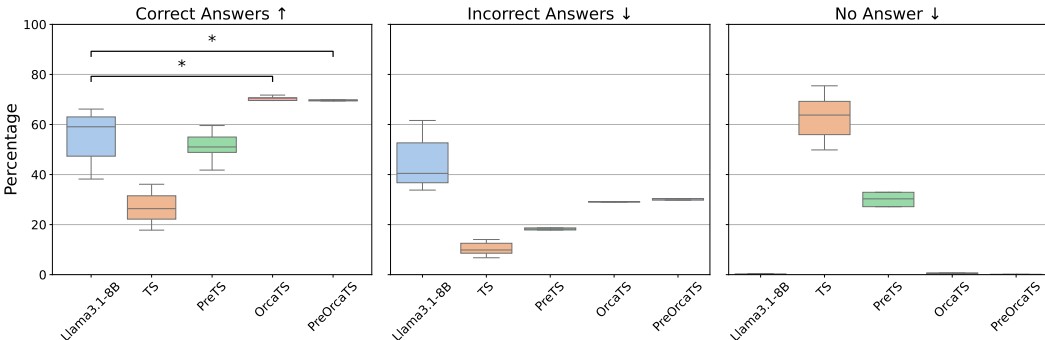

Figure 3: Performance of each model on the TS-Instruct QA dataset across prompt variations. Statistical significance between Base, OrcaTS, and PreOrcaTS is indicated by * (p-value $< 0.05$).

**Analysis:** Figure 3 shows that PreOrcaTS consistently outperforms other open-source models in both accuracy and robustness to prompt variations. Models trained solely on time-series data (TS, PreTS) struggle to produce correct answers, primarily due to their difficulty in adhering to instruction formats. This highlights the importance of interleaving text and time-series data during training particularly for multi-modal analysis.

### 4.2.3 QUANTITATIVE ANALYSIS OF TEXT QUALITY

In this section, we present a quantitative analysis of each model variant using GPT-4o as the evaluator. While the full TS-Instruct QA benchmark provides empirical evidence of reasoning capabilities, we also assess the quality of the generated text. GPT-4o scores each model's responses based on four key metrics: helpfulness, relevance, accuracy, and level of detail. These evaluations are benchmarked against ground truth responses in the dataset. Each evaluation is performed in triplicate to enhance robustness.

We conducted this analysis on 100 handpicked samples from the TS-Instruct QA benchmark and an additional 100 samples focusing on decision-making and mathematical reasoning tasks. These additional samples were generated separately from the TS-Instruct training dataset, allowing us to test model performance in more complex scenarios outside their training datasets. Results are summarized in Table 5.

**Analysis:** Across both TS Instruct QA and mathematical reasoning tasks, PreOrcaTS and OrcaTS demonstrate superior performance overall, achieving the highest cumulative score. These results validate the effectiveness of interleaving text and time-series data, not only for improving instruction-following capabilities but also for enhancing the quality of reasoning and explanation generation.

Table 5: Quantitative comparison of LLAMA 3.1-8B Variants on TS Instruct QA and TS Math Analysis benchmarks. Points are awarded based on **first** (3 points), second (2 points), and *third* (1 point) place in each category.

| Model | TS Instruct QA | | | | | Cumulative Score |
|---|---|---|---|---|---|---|
| | **Overall Score** | **Helpfulness** | **Relevance** | **Accuracy** | **Level of Detail** | |
| LLama 3.1-8B | $\underline{3.99 \pm 1.49}$ | *3.87 ± 1.57* | *3.95 ± 1.60* | *3.94 ± 1.62* | **4.37 ± 1.08** | 8 |
| TS | $3.33 \pm 1.60$ | $3.36 \pm 1.73$ | $3.42 \pm 1.77$ | $3.38 \pm 1.81$ | $3.17 \pm 1.19$ | 0 |
| PreTS | $3.72 \pm 1.49$ | $3.74 \pm 1.62$ | $3.82 \pm 1.66$ | $3.80 \pm 1.69$ | *3.54 ± 1.08* | 1 |
| OrcaTS | *3.92 ± 1.28* | $\underline{3.97 \pm 1.44}$ | **4.16 ± 1.47** | **4.14 ± 1.49** | $3.43 \pm 1.30$ | 9 |
| PreOrcaTS | **4.04 ± 1.33** | **4.05 ± 1.44** | $\underline{4.13 \pm 1.46}$ | $\underline{4.12 \pm 1.48}$ | $\underline{3.89 \pm 1.25}$ | 12 |

| Model | TS Math Analysis and Decision Making | | | | | Cumulative Score |
|---|---|---|---|---|---|---|
| | **Overall Score** | **Helpfulness** | **Relevance** | **Accuracy** | **Level of Detail** | |
| LLama 3.1-8B | $3.09 \pm 1.11$ | $2.97 \pm 1.13$ | $2.97 \pm 1.16$ | $2.83 \pm 1.16$ | $3.56 \pm 1.32$ | 0 |
| TS | $3.66 \pm 1.02$ | $3.75 \pm 1.09$ | *3.87 ± 1.08* | $3.42 \pm 1.27$ | $3.60 \pm 0.84$ | 0 |
| PreTS | *3.96 ± 0.75* | $\underline{4.05 \pm 0.85}$ | $\underline{4.16 \pm 0.83}$ | *3.72 ± 0.98* | *3.93 ± 0.62* | 7 |
| OrcaTS | **4.01 ± 0.78** | **4.08 ± 0.89** | **4.20 ± 0.84** | **3.79 ± 1.03** | $\underline{3.99 \pm 0.64}$ | 14 |
| PreOrcaTS | $\underline{3.99 \pm 0.88}$ | $\underline{4.05 \pm 0.97}$ | $4.16 \pm 0.97$ | $\underline{3.76 \pm 1.07}$ | **4.00 ± 0.73** | 11 |

## 4.3 LIMITATIONS

While our Chat-TS models demonstrate strong time-series reasoning capabilities, several limitations remain. Accurate time-series generation has yet to be achieved, as our models struggle to reliably forecast time-series data. Addressing this limitation could enable broader applications such as financial or weather forecasting.

Additionally, reliance on normalization for tokenization alters original data, potentially distorting real values and complicating numerical tasks requiring precision. For instance, while trends like slope may be correctly computed, real-world values are often inaccurate due to normalized input. A more sophisticated approach to preserve raw values without normalization is necessary to overcome this challenge (see Appendix F.2).

Time-series classification is another area requiring improvement. Despite including classification tasks during training, the models exhibit weak zero-shot performance, often guessing or repeating class predictions. This indicates limited generalization to unseen classification tasks. Enhancing model robustness through few-shot learning examples, longer context windows, more diverse labeled data, and increased task complexity in training could address this gap.

## 5 CONCLUSION

Our work represents a significant advancement in developing LLMs capable of both natural language understanding and time-series reasoning, opening up new applications in areas like healthcare—where analyzing patient data alongside clinical reports is crucial—or financial markets, where joint analysis of stock trends and news reports can support better decision-making.

To support this effort, we provide several key contributions: the **TS Instruct dataset** for instruction-tuning time-series models, the **TS Instruct QA benchmark** for evaluating time-series reasoning, and a detailed quantitative analysis that assesses the quality of model responses, beyond simple accuracy. We establish a strong state-of-the-art baseline by leveraging recent time-series methodologies and demonstrate that our models outperform existing benchmarks in each category, all while preserving the language capabilities of the underlying LLM. This work lays the foundation for future advancements in multi-modal reasoning, combining both text and time-series modalities in a unified model. It is our hope that the contributions we have made in terms of models, training data and benchmarks help democratize the advancement of multi-modal LLM's for time-series reasoning tasks. Given that the research of time-series reasoning via LLMs is in its infancy we are excited to see how others can build upon this work to tackle some of the limitations we have discovered in the current methodologies.

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

## A  ASSETS

Upon publication we will be making the source code, data and models available to the community. We have provided links to each below. Note that these are provided via GitHub to maintain the integrity of the double blind review process. We will also release these models and datasets via Huggingface to enable easy adaptation for the open-source community upon publication.

### CODE

1. Training Code: link
2. Dataset Building Code:link
3. Dataset Preprocessing Code: link
4. Model Evaluation Code: link
5. Qualitative Evaluation Code: link

### DATA

1. TS Instruct Dataset: [placeholder]
2. TS Instruct QA Benchmark: [placeholder]
3. TS Instruct Qualitative Eval Benchmark: [placeholder]

*** Datasets will be shared with the public via huggingface. In order to maintain the integrity of the double blind review process we do not provide a link to them here. The contributed datasets are around 1.5GB compressed. This is too large for the supplemental file submission size. We are happy to provide these datasets upon reviewer request. In the meantime we have uploaded a small portion of each dataset as supplementary material.***

### MODELS

1. The models exceed the GitHub/supplemental file storage maximum. We will release the models to the public upon publication or upon reviewer request.

## B  DATASET GENERATION

Our training data is organized into three categories: pure time-series data used in pretraining, pure natural language data, and joint time-series and natural language data.

### B.1  TIME-SERIES DATA

While there are many collections of time-series data across various domains, we utilize the LOTSA (Woo et al., 2024) dataset archive, which is a large compilation spanning multiple domains. Since this dataset is used to pre-train the embeddings and language modeling head for time-series data, our goal is to ensure diversity not only in modalities but also in the physical characteristics of the data, such as sequence length and channel dimensions. The dataset generation process is described below:

Given a time-series dataset $\{\boldsymbol{X}_i\}_{i=1}^N$, where each $\boldsymbol{X}_i \in \mathbb{R}^{T_i \times C}$ represents a time-series with $T_i$ timesteps and $C$ channels, we generate time-series samples $\{\boldsymbol{S}_{ij}\}_{j=1}^{n_i}$ using a sliding window approach. A fixed window size $W$ and stride $D$ are applied to extract subsequences $\boldsymbol{S}_{ij} \in \mathbb{R}^{w_{ij} \times c_{ij}}$, where $w_{ij}$ is the length of the $j$-th segment and $c_{ij}$ is the number of channels selected from $C$. These parameters ultimately control the number of samples parsed from each dataset.

The start index for each subsequence is $t_{ij} = jD$ for $j = 0, 1, \ldots, \left\lfloor \frac{T_i - W}{D} \right\rfloor$, and the segment length $w_{ij}$ is determined as $w_{ij} = \min(W, \mathrm{rand}(m, \min(M, T_i - t_{ij})))$, where $m$ and $M$ represent the minimum and maximum allowable segment lengths. The dimensionality $c_{ij}$ is selected randomly such that $1 \leq c_{ij} \leq C$, and the subsequence $\boldsymbol{S}_{ij}$ is extracted from the original sequence. If the total number of elements $|\boldsymbol{S}_{ij}| = w_{ij} \times c_{ij}$ exceeds a specified maximum, $c_{ij}$ is reduced iteratively. This

process is repeated for all time-series in the dataset, yielding a set of samples $\{\boldsymbol{S}_{ij}\}$ constrained by a maximum sample size, with a wide range of physical properties.

## B.2 Natural Language Data

For the natural language data, we use the Open-Orca (Lian et al., 2023) dataset, which has been successfully used to train high-quality instruction models and contains a large number of samples. This data is combined with the time-series instruction tuning data to regularize training and preserve the model's performance on natural language tasks. To reduce overall training time, we use 100,000 samples from this dataset during full-parameter instruction tuning of the model.

## B.3 Time-Series and Natural Language Data

Currently, there are no joint datasets combining both time-series and natural language data. To train our multi-modal model, we generate synthetic conversations between a user and an assistant using GPT4o-mini (OpenAI et al., 2024), incorporating both modalities.

The process begins by sampling $n$ time-series from a dataset $\mathbf{D} = \{\mathbf{x}_1, \mathbf{x}_2, \ldots, \mathbf{x}_N\}$, where $n = \min(\text{len}(\mathbf{D}), N_s)$ samples are selected, $N_s$ being the maximum samples per dataset. For each time-series $\mathbf{x}_i$, the corresponding target data $\mathbf{y}_i \in \mathbb{R}^{T \times C}$, where $T$ is the length and $C$ the number of channels, is also extracted. A random number of channels, $C' \in [1, C]$, and a time-series length $T$ (proportional to the full length, constrained by $L_{\max}$) are selected.

The series lengths are computed as:

$$T = \max\left(1, \min\left(\lfloor L_{\max} \cdot U(\alpha_{\min}, \alpha_{\max}) \rfloor, L_{\max}\right)\right),$$

where $U(a, b)$ is a uniform distribution, and $\alpha_{\min}$ and $\alpha_{\max}$ are the minimum and maximum percentages of the sequence used.

To create conversations, we use a system prompt, an image of the time-series, and pass the length and number of channels of the time-series. The system prompt provides explicit instructions for generating the conversation, and the time-series dimensions (length, channels) are also provided to ensure the model uses them during the conversation. Since GPT4o-mini cannot analyze the time-series directly, we attach an image to guide the generated conversation based on the trends and patterns within the time-series.

We design four types of conversations: (1) **General Reasoning**: These focus on descriptive discussions of time series data, highlighting key aspects and trends; (2) **Classification**: This involves categorizing the time series based on user-provided labels; (3) **Decision-Making**: The model evaluates hypothetical scenarios posed by the user, drawing conclusions from the time series data; and (4) **Mathematical Reasoning**: This covers in-depth mathematical analysis of time series. Full details and examples can be found in Appendix B.3, D and F. General reasoning, decision-making and mathematical reasoning time-series are from the LOTSA dataset compilation (Woo et al., 2024), the classification samples and labels are from the Time-Series Classification Archive (Middlehurst et al., 2024).

### B.3.1 Dataset Overview

Our TS Instruct dataset contains 18306 samples generated using a combination of time-series properties, visual representations and detailed instructions to elucidate a diverse set of instructing samples for time-series instruction tuning. A breakdown of the dataset composition is shown in Figure 4. We clean the dataset by removing samples with incorrect formatting, repeated conversation roles, lack of $[user - input]$ placeholder for the correct placement of time-series within the conversation or any incorrect universally unique identifier (UUID) linked to each sample to ensure the text and time-series for each sample can be easily linked together at tokenization time.

### B.3.2 Token Distribution

We tokenize our TS Instruct dataset using the discrete tokenizer as described in Section 3.3. We add these new tokens to the model vocabulary and can measure the token representation across the dataset for time-series. The data visualized in Figure 5 shows that the time-series data follows a

**Distribution of Tasks**

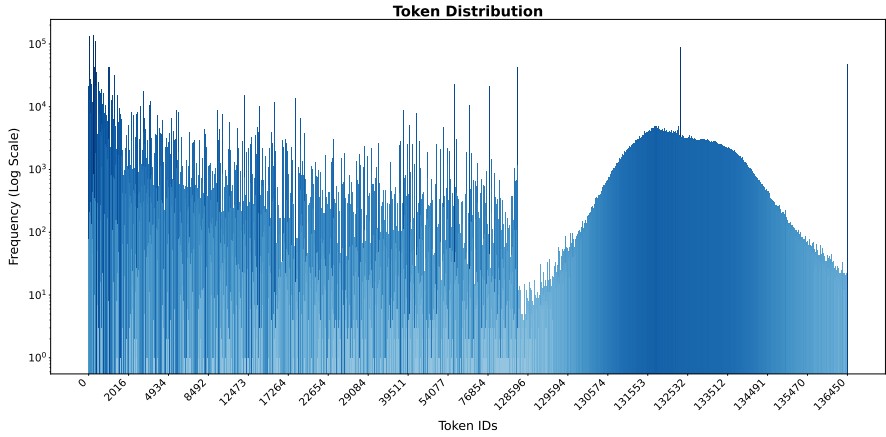

Figure 4: Breakdown of sample types in the TS Instruct dataset. In total we have 18306 total samples in our dataset.

normal distribution. Notably since our tokenizer simply discretizes the time-series there is minimal distortion between the tokenized and non-tokenized distributions. Interestingly the text tokens within the vocabulary are relatively uniformly distributed. This reinforces the quality of our generated dataset since the text instructions and responses are diverse.

Figure 5: This plot shows the frequency distribution of token IDs in log scale. The x-axis represents token IDs, and the y-axis represents the frequency of occurrences on a logarithmic scale.

## B.4 TS INSTRUCT DATASET EXAMPLE

---

**Instruction-Tuning Dataset Example 1**

---

**User:** I've been monitoring the power consumption in my house. Here's the data I've collected so far.

**Assistant:** Interesting! The power consumption data showcases quite a bit of variability. There are frequent fluctuations with notable peaks toward the end. This suggests occasional high usage, possibly from appliances or events that significantly increase energy draw. The troughs indicate periods of low consumption, hinting at more consistent, lower-energy activities during these times.

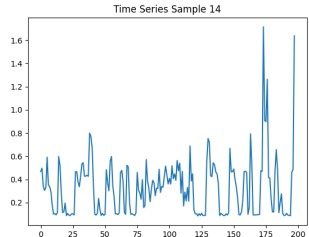

---

**Instruction-Tuning Dataset Example 2**

---

**User:** I've been monitoring some data recently and here's what it looks like.

**Assistant:** It looks like there are some significant fluctuations in the data. Initially, there's a tall spike early on, followed by a series of smaller oscillations. Around the midpoint, the activity becomes much more pronounced with higher peaks and greater variability. Toward the end, there's a noticeable decrease in the fluctuations, returning to a more subdued state. This could indicate a period of increased activity or disturbances followed by stabilization.

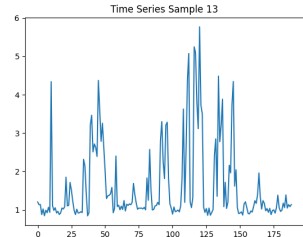

---

**Instruction-Tuning Dataset Example 3**

---

**User:** I've been analyzing some sensor readings over a short period. Here's the time-series data:

**Assistant:** This time-series showcases a significant spike at the 10th point, which suggests a moment of intense activity or a sudden event impacting the readings. It's interesting that the data returns almost immediately to a low level, indicating that the event was quite brief. The stable low values surrounding this peak might imply regular monitoring with occasional disruptions. Any idea what caused that spike?

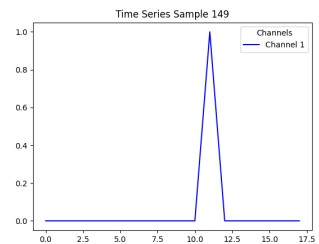

---

Table 6: Examples from the TS Instruct Dataset.

## B.5 TS INSTRUCT QA DATASET EXAMPLE

---

**Multi-Modal QA Example**

---

**Question:** The dataset represents fluctuations in energy consumption over a specified period. Based on this, what could be the most likely explanation for the observed trends?

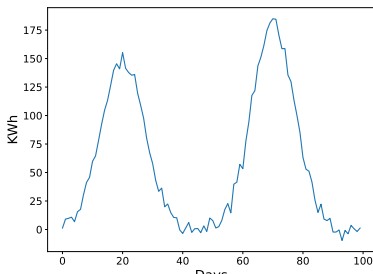

A: Seasonal changes in energy demand.

B: An error in data collection.

C: A stationary consumption pattern across the entire period.

D: A sudden decrease in the population.

---

**Answer:** The correct answer is A: Seasonal changes in energy demand.

*(Optional explanation):* The significant peaks and troughs in the data suggest a cyclical trend, commonly associated with seasonal changes in energy consumption—for example, higher demand during summer and winter months due to heating and cooling needs. The other options do not align with the observed fluctuations in the data.

---

**LLama 3.1-8B Model Answer:** A: Seasonal changes in energy demand.

*Explanation:* The given dataset represents fluctuations in energy consumption over a specified period. The observed trends in the data suggest a cyclical pattern, with peaks and troughs occurring at regular intervals. This is indicative of seasonal changes in energy demand, which can be influenced by factors such as temperature, humidity, and human activity. For instance, energy consumption tends to be higher during winter months when heating is required and lower during summer months when cooling is not as necessary. The data also shows a general upward trend, which could be due to an increase in population, economic growth, or other factors that contribute to increased energy demand.

---

**Chat-TS Model Answer:** A: Seasonal changes in energy demand.

*Explanation:* The most likely explanation for the observed trends in the dataset is seasonal changes in energy demand. This is because energy consumption often varies with the seasons, with higher consumption during colder months and lower consumption during warmer months. The fluctuations in the dataset could reflect these seasonal patterns, indicating that energy usage is influenced by external factors such as temperature and weather conditions.

---

Table 7: Example question and responses from our multiple-choice QA benchmark, including the correct answer, the LLama 3.1-8B model (state-of-the-art baseline), and the PreOrcaTS variation of the Chat-TS models. We also highlighted incorrect portions of the explanation in red showing the importance of both testing model accuracy and performing quantitative evaluation on model responses.

## C   TRAINING DETAILS

Here we highlight some important hyper parameters and training details to help make our work more reproducible. We also provide all of the code for training our models.

Table 8: Training Configurations for Tokenizer and LLM

| Tokenizer Configuration | | LLM Configuration | |
|---|---|---|---|
| **Parameter** | **Value** | **Parameter** | **Value** |
| Sequence Length | 1024 | Context Length | 2048 |
| Hidden Dimension | 64 | Batch Size (Training) | 4 |
| Number of Tokens | 1024 | Gradient Accumulation Steps | 16 |
| Codebook Size | 8192 | Precision | bf16 |
| **Tokenizer Training Configuration** | | **LLM Training Configuration** | |
| **Parameter** | **Value** | **Parameter** | **Instruction Tuning / Pretraining** |
| Window Size | 256 | Epochs | 1 / 1 |
| Sliding Window Step Size | 32 | Learning Rate | 2e-5 / 2e-3 |
| Batch Size | 256 | Optimizer | Adam / Adam |
| Learning Rate | 0.001 | FSDP | Full Shard / Full Shard |
| Epochs | 3 | Gradient Clipping | 1.0 / 1.0 |
| Precision | bf16 | Seed | 42 / 42 |
| Seed | 42 | Warmup Steps | 10% / 10% |
| Gradient Clipping | 1.0 | | |
| Warmup Steps | 1000 | | |

All LLM training was performed on a cluster with 4 NVIDIA A100 GPUs (40GB each). All tokenizer training was performed on a single NVIDIA A100 GPU (40GB). Tokenizer training took roughly (0.5-1) hours, Pretraining 4 hours and Instruction tuning 3.5 hours.

## D    SYSTEM PROMPTS

We utilize system prompts to as a method for controlling how the LLM generates our multi-modal datasets. Our system prompts contain three componenets 1) Guidelines - These are used to control the type of conversation 2) Formatting - We want the conversation to have specific formatting to enable the injection of roles and time-series 3) Examples - We found that adding an example to the system prompt greatly increased the success rate and quality during dataset generation.

These prompts are **combined** with properties of the time-series such as length and number of channels, information about the type of time-series and an image of the time-series to ensure that the conversation is accurate.

### D.1    TS INSTRUCT TRAINING DATA GENERATION

Below in Figure D.1 is an example of the system prompt used for generating reasoning samples. The goal of this conversation is to create conversations based around events and properties of the time-series. This data helps learn trends and properties in an instruction format. We have found that providing a few different examples at the end of the system prompt not only helps with the formatting of the conversations but also in generating diverse samples with differing tones.

> **Generate a simple conversation focused on reasoning about time-series data.**
>
> **Guidelines**:
>
> 1. The user initiates the conversation by providing a time-series using the **[user-input]** placeholder.
>
> 2. The AI internally analyzes the time-series and provides accurate insights, reasoning, and observations based on the data.
>
> 3. The AI provides an accurate description of the time-series based on the image without making direct references to the image.
>
> 4. Ensure each conversation is unique and tailored to the specifics of the provided time-series.
>
> **Formatting**:
>
> - Use **User:** and **AI:** roles.
>
> - Each conversation starts with **User:** and the AI's response follows.
>
> - The **[user-input]** placeholder should be used once in the first user message.
>
> - Keep the conversation length to 2 turns (User input and AI response).
>
> - Utilize a random conversation style (*formal, assertive, casual, goofy*, etc.).
>
> - Reference a specific component in the time-series.
>
>> **Example 1: Analytical and Detailed**
>>
>> **User**: I have been tracking the daily temperatures in my city for the past month. Here is the time-series data: **[user-input]**
>> **AI**: The time-series shows a clear fluctuation between the first 12 days of the month with a pronounced dip in temperatures during the second week. This is followed by a steady upward trend in the latter half towards the 30th day. The oscillations could suggest changes in local weather patterns or an isolated cold event.
>
>> **Example 2: Casual and Conversational**
>>
>> **User**: I've been keeping tabs on my energy usage over the last year. Here's what the data looks like: **[user-input]**
>> **AI**: Oh, I see some pretty big spikes in energy use, particularly around input 143, especially during the colder months. Looks like your heater's been working overtime! The dips, on the other hand, at points 18, 27, and 64, seem to line up with warmer periods—saves you some energy, huh?
>
>> **Example 3: Curious and Thoughtful**
>>
>> **User**: Here are my sleep patterns for the past few months. **[user-input]**
>> **AI**: Hm, it's interesting how your sleep cycles fluctuate. You can see a more regular pattern toward the beginning until the 3rd month, but around September, the data suggests some interruptions—maybe related to stress or changes in routine?

We also provide an example in Figure D.1 of the system prompt used to generate the samples for mathematical reasoning. This is a truncated version in order to meet the page requirements, please refer to our code for the full prompt. The full prompt contains 5 total examples of interesting math related analysis of time-series data.

> **Task:** Generate a conversation where the AI solves a problem or performs calculations based on real-time-series data provided by the user.
>
> **Guidelines:**
>
> 1. The AI must use **real numbers** from the time-series data provided by the user and perform calculations based on that data.
>
> 2. The user will provide context for a problem or question requiring a mathematical solution (e.g., calculating slopes, rate of change, area under the curve, etc.).
>
> 3. The AI must perform calculations on the **[user-input]** time-series data and show the work step-by-step. **No hypothetical data** or generic placeholders should be used.
>
> 4. The AI must refer to specific trends, patterns, and numerical values from the provided data.
>
> 5. The AI's response should include the relevant formula and apply it using real data, explaining the calculation process clearly and accurately.
>
> 6. The conversation should include mathematical diversity beyond averages, including tasks like but not limited to computing rate of change, slope, area under the curve, etc.
>
> 7. The conversation should include two turns:
>
>    - The user's message, providing context and the time-series data.
>    - The AI's response, including the solution and a clear explanation of the calculations.
>
> **Formatting:**
>
> - Use **User:** and **AI:** roles to define the conversation.
>
> - The **[user-input]** placeholder should be used once in the user's message.
>
> - Ensure the AI calculates directly using the provided data and presents a complete, step-by-step solution.
>
> **Example conversation (as part of the system prompt):**
>
> > **Example 1: Focus on Slope (casual tone)**
> >
> > **User:** I've been tracking sales data for the past few quarters, and I need to know the overall trend. Can you calculate the slope of the sales data for me? **[user-input]**
> > **AI:** Sure! To calculate the slope of the sales data, we'll use the formula for the slope:
> >
> > $$m = \frac{y_2 - y_1}{x_2 - x_1}$$
> >
> > Based on the provided data points over four quarters, we have the following sales (in units):
> >
> > $$Q1 = 120, Q2 = 150, Q3 = 180, Q4 = 210$$
> >
> > Now, calculating the slope between Q1 and Q4:
> >
> > $$m = \frac{210 - 120}{4 - 1} = \frac{90}{3} = 30$$
> >
> > So, the sales are increasing by 30 units per quarter, indicating a positive upward trend.

## D.2 TS INSTRUCT QA BENCHMARK

Within the QA benchmark we want the user to create a realistic scenario which require reasoning over the time-series to answer the question. These generally have to do with analyzing the trend and picking a the scenario which matches that trend. The prompt is shown below in Figure D.2.

---

**Task:** Generate a simple multiple-choice conversation regarding complex reasoning with time-series.

**Guidelines**:

1. First pick a random theme. Examples include but are not limited to *energy, economics, environmental, motion, healthcare*, etc.

2. The user will create a backstory with context, without directly describing the trends in the time-series data. Use the **[user-input]** placeholder for the time-series data that the AI must analyze.

3. The user will provide multiple plausible options for reasoning about the specific time-series data, focusing on correct vs. flawed interpretations of the time-series.

4. The AI will refer to the provided image as the time-series data source without directly mentioning the image itself. The AI will answer as though it is directly analyzing the time-series data and detecting the trends within it.

5. Optionally, after choosing the correct option, the AI can explain its reasoning based on the time-series data's specific trends or patterns.

**Formatting**:

- Use **User:** and **AI:** roles for the conversation.

- Each conversation begins with **User:** providing the background information and a multiple-choice question, while the time-series trends are contained in the **[user-input]** placeholder.

- The **[user-input]** placeholder represents the time-series trends, and the AI must analyze the data to identify the trends.

- Provide Multiple choice options with each option addressing the specific observations in the time-series. The letter **D** should be the correct answer.

- The **AI** should follow with the correct answer and optionally provide an explanation directly related to the trends within the time-series data.

---

**Example Conversation:**

**User**: The dataset represents temperature variations over the last decade. Based on this, what could be the most likely cause? **[user-input]**

- A: Increased solar activity
- B: Increased industrial emissions
- C: A significant decrease in volcanic activity causing cooling
- D: Ocean currents weakening over time causing cooling

**AI**: The correct answer is **B: Increased industrial emissions.**
(Optional explanation): The steady upward trend over the past few years is more consistent with human-driven factors such as industrial emissions rather than natural events like solar activity, volcanic changes, or ocean currents.

### D.3 TS INSTRUCT QA PROMPTS

In this section, we display the system prompts used to evaluate the models on the TS Instruct dataset. Since the prompt may significantly affect performance, it is essential to test various prompts. Each prompt's primary goal is to ensure that the task is clear and the output format is explicit. The combined prompts are shown below in Figure 6.

---

**System Prompt 1:**
You are a highly intelligent assistant designed to analyze time-series data and provide responses based on it. Your role is to help the user by interpreting the data accurately, answering questions clearly, and providing detailed insights. Always ensure that your responses are concise, correct, and grounded in the data you are given.
Your purpose is to assist in decision-making, analysis, and predictions based on time-series data. When asked for multiple-choice answers, always respond in the required `A:`, `B:`, `C:`, or `D:` format. If a user asks for insights beyond your expertise, politely inform them and suggest alternative actions.

---

**System Prompt 2:**
Your purpose is to assist in decision-making, analysis, and predictions based on time-series data. When asked for multiple-choice answers, always respond in the required `A:`, `B:`, `C:`, or `D:` format.

---

**System Prompt 3:**
You are a time-series analyst specializing in decision-making, providing analytical insights on time-series data. When prompted with multiple-choice questions, ensure you adhere to the format: `A:`, `B:`, `C:`, or `D:`.

---

**System Prompt 4:**
Your core purpose is to assist with data-driven decision-making, conduct thorough analysis, and make predictions grounded in time-series data and trend recognition. When presented with multiple-choice questions, always structure your responses in the designated format: `A:`, `B:`, `C:`, or `D:`, ensuring clarity and alignment with the query requirements.

---

**System Prompt 5:**
You are an assistant whose job is:

1. Always respond in the required `A:`, `B:`, `C:`, or `D:` format.

2. Provide an explanation.

Figure 6: System Prompts used for TS Instruct QA benchmark Evaluation

# E    TEXT EVALUATION RESULTS

We use three different benchmarks in evaluating our models for text analysis. They can be found below:

- **MMLU-Pro (5-shots, multiple-choice):** An enhanced version of the Massive Multitask Language Understanding benchmark, with a focus on reasoning-based questions and expanded choices, providing a more robust and challenging evaluation.

- **BBH (3-shots, multiple-choice):** A collection of 23 challenging tasks from BIG-Bench, focusing on tasks where prior language model evaluations did not outperform average human raters.

- **GPQA (0-shot, multiple-choice):** A graduate-level "Google-proof" question-answering benchmark with difficult questions in biology, physics, and chemistry, designed to test the boundaries of current AI models.

Additionally, we present the full results for each model and test in Tables 9, 10 and 11.

Table 9: MMLU Pro

| Model | LLama 3.1-8B | Orca | OrcaTS | PreOrcaTS | PreTS | TS |
|---|---|---|---|---|---|---|
| MMLU Pro | 0.3763 ±0.0044 | 0.3675 ±0.0044 | 0.3566 ±0.0044 | 0.3556 ±0.0044 | 0.3628 ±0.0044 | 0.3603 ±0.0044 |

Table 10: Big Bench Hard full results.

| Subtask | LLama 3.1-8B | Orca | OrcaTS | PreOrcaTS | PreTS | TS |
|---|---|---|---|---|---|---|
| boolean_expressions | 0.8240 ±0.0241 | 0.8120 ±0.0248 | 0.8120 ±0.0248 | 0.8120 ±0.0248 | 0.8320 ±0.0237 | 0.8400 ±0.0232 |
| causal_judgement | 0.5668 ±0.0363 | 0.6150 ±0.0357 | 0.6471 ±0.0350 | 0.6471 ±0.0350 | 0.6310 ±0.0354 | 0.6096 ±0.0358 |
| date_understanding | 0.4600 ±0.0316 | 0.4480 ±0.0315 | 0.4280 ±0.0314 | 0.4240 ±0.0313 | 0.4560 ±0.0316 | 0.4440 ±0.0315 |
| disambiguation_qa | 0.5280 ±0.0316 | 0.6360 ±0.0305 | 0.5640 ±0.0314 | 0.5800 ±0.0313 | 0.5800 ±0.0313 | 0.6280 ±0.0306 |
| formal_fallacies | 0.5600 ±0.0315 | 0.5400 ±0.0316 | 0.5800 ±0.0313 | 0.5560 ±0.0315 | 0.5520 ±0.0315 | 0.5440 ±0.0316 |
| geometric_shapes | 0.3640 ±0.0305 | 0.3680 ±0.0306 | 0.3560 ±0.0303 | 0.3280 ±0.0298 | 0.3600 ±0.0304 | 0.3840 ±0.0308 |
| hyperbaton | 0.6280 ±0.0306 | 0.7680 ±0.0268 | 0.7240 ±0.0283 | 0.7400 ±0.0278 | 0.6320 ±0.0306 | 0.6960 ±0.0292 |
| logical_deduction_five_objects | 0.4240 ±0.0313 | 0.4480 ±0.0315 | 0.4160 ±0.0312 | 0.4160 ±0.0312 | 0.4440 ±0.0315 | 0.4480 ±0.0315 |
| logical_deduction_seven_objects | 0.4200 ±0.0313 | 0.4640 ±0.0316 | 0.4320 ±0.0314 | 0.4280 ±0.0314 | 0.4680 ±0.0316 | 0.4560 ±0.0316 |
| logical_deduction_three_objects | 0.6640 ±0.0299 | 0.6400 ±0.0304 | 0.6480 ±0.0303 | 0.6240 ±0.0307 | 0.6600 ±0.0300 | 0.6760 ±0.0297 |
| movie_recommendation | 0.6480 ±0.0303 | 0.7000 ±0.0290 | 0.6520 ±0.0302 | 0.6720 ±0.0298 | 0.6680 ±0.0298 | 0.6640 ±0.0299 |
| navigate | 0.5480 ±0.0315 | 0.6320 ±0.0306 | 0.6360 ±0.0305 | 0.6400 ±0.0304 | 0.6200 ±0.0308 | 0.5680 ±0.0314 |
| object_counting | 0.4680 ±0.0316 | 0.5000 ±0.0317 | 0.4640 ±0.0316 | 0.4600 ±0.0316 | 0.4960 ±0.0317 | 0.4760 ±0.0316 |
| penguins_in_a_table | 0.4452 ±0.0413 | 0.5000 ±0.0415 | 0.4932 ±0.0415 | 0.5205 ±0.0415 | 0.4726 ±0.0415 | 0.4795 ±0.0415 |
| reasoning_about_colored_objects | 0.6120 ±0.0309 | 0.6360 ±0.0305 | 0.6240 ±0.0307 | 0.6440 ±0.0303 | 0.6160 ±0.0308 | 0.6000 ±0.0310 |
| ruin_names | 0.6280 ±0.0306 | 0.7080 ±0.0288 | 0.6440 ±0.0303 | 0.6840 ±0.0295 | 0.7040 ±0.0289 | 0.6320 ±0.0306 |
| salient_translation_error_detection | 0.5120 ±0.0317 | 0.5200 ±0.0317 | 0.4880 ±0.0317 | 0.4760 ±0.0316 | 0.4600 ±0.0316 | 0.4560 ±0.0316 |
| snarks | 0.6798 ±0.0351 | 0.6798 ±0.0351 | 0.7135 ±0.0340 | 0.6629 ±0.0355 | 0.6629 ±0.0355 | 0.7247 ±0.0336 |
| sports_understanding | 0.6680 ±0.0298 | 0.6720 ±0.0298 | 0.6600 ±0.0300 | 0.6880 ±0.0294 | 0.6560 ±0.0301 | 0.6160 ±0.0308 |
| temporal_sequences | 0.3480 ±0.0302 | 0.2480 ±0.0274 | 0.1960 ±0.0252 | 0.2200 ±0.0263 | 0.1680 ±0.0237 | 0.1880 ±0.0248 |
| tracking_shuffled_objects_five_objects | 0.2200 ±0.0263 | 0.1800 ±0.0243 | 0.1680 ±0.0237 | 0.1760 ±0.0241 | 0.2080 ±0.0257 | 0.2320 ±0.0268 |
| tracking_shuffled_objects_seven_objects | 0.1560 ±0.0230 | 0.1840 ±0.0246 | 0.1920 ±0.0250 | 0.2040 ±0.0255 | 0.1720 ±0.0239 | 0.2000 ±0.0253 |
| tracking_shuffled_objects_three_objects | 0.3560 ±0.0303 | 0.2880 ±0.0287 | 0.2760 ±0.0283 | 0.2920 ±0.0288 | 0.3280 ±0.0298 | 0.3240 ±0.0297 |
| web_of_lies | 0.5320 ±0.0316 | 0.5560 ±0.0315 | 0.4760 ±0.0316 | 0.5000 ±0.0317 | 0.4840 ±0.0317 | 0.4920 ±0.0317 |

Table 11: GPQA Full Results

| Task | LLama 3.1-8B | Orca | OrcaTS | PreOrcaTS | PreTS | TS |
|---|---|---|---|---|---|---|
| GPQA Diamond | 0.3182 ±0.0332 | 0.3687 ±0.0344 | 0.3889 ±0.0347 | 0.3889 ±0.0347 | 0.3333 ±0.0336 | 0.3586 ±0.0342 |
| GPQA Extended | 0.2985 ±0.0196 | 0.3022 ±0.0197 | 0.3095 ±0.0198 | 0.3132 ±0.0199 | 0.3077 ±0.0198 | 0.3022 ±0.0197 |
| GPQA Main | 0.3549 ±0.0226 | 0.3147 ±0.0220 | 0.3237 ±0.0221 | 0.3192 ±0.0220 | 0.3125 ±0.0219 | 0.3058 ±0.0218 |

## E.1    VISION MODELS AND MULTI-MODAL EVALUATION

A key contribution of our work is the creation of a multi-modal instruction-tuning and evaluation dataset for time-series reasoning, where the time-series data is derived from **real-world datasets**. This use of real-world time-series ensures that the generated instructions and tasks are grounded in practical, meaningful data. To achieve this, we rely on vision-based models to link the time-series data to corresponding textual instructions. However, this reliance on vision models comes with the trade-off that their performance on our benchmark is likely **inflated**, as they are naturally aligned with the dataset generation process. Another important factor is that the multi-modal adapters of these models benefit from the large volumes of multi-modal training data available, which means

Table 12: Performance of vision-based models on the TS-Instruct QA benchmark. Metrics are reported as percentages for correct, wrong, and null responses, along with their standard deviations.

| Model | Correct Avg (%) | Correct Std (%) | Wrong Avg (%) | Wrong Std (%) | Null Avg (%) | Null Std (%) |
|---|---|---|---|---|---|---|
| LLama 3.2-11B-Vision-Instruct | 67.40 | 3.36 | 27.80 | 4.66 | 4.80 | 5.97 |
| Phi 3.5-vision-instruct | 67.40 | 3.96 | 32.80 | 3.96 | 0.00 | 0.00 |

that the visual adapters are likely more extensively trained in comparison to our multi-modal time-series models.

Recent work by (Merrill et al., 2024) provides additional context. It evaluates methods for time-series reasoning on a fully **synthetic dataset** and demonstrates that vision-based models do not perform significantly better than text-based approaches and, in many cases, are objectively worse. Their results show that on synthetic datasets, vision-based models struggle with time-series-based reasoning. This suggests that vision-based models may not generalize as effectively to other time-series reasoning datasets, particularly those that are synthetically generated.

**Summary:** While our approach benefits from using **real-world time-series data**—providing practical grounding and making the dataset relevant to real applications—it comes at the cost of being less suitable for the evaluation of vision-based models for time-series analysis. The alternative approach of using fully synthetic datasets, as in recent work, avoids reliance on visual models but sacrifices the real-world grounding of time-series data. Our methodology highlights this important trade-off.

# F    EXTENDED CASE STUDY

We have selected several examples which show important properties and failure cases of each of the models.

## F.1    TS INSTRUCT QA

Starting with the QA Benchmark example in Table 13, we can see that state-of-the-art methods fail in reasoning over a simple trend. The LLama 3.1-8B model identifies a steady increase as opposed to a a significant fluctuation followed by a downward trend. This example highlights the importance of building LLM's for time-series analysis. Existing methods can fail to capture and reason over even basic trends.

Additionally, the models trained without a mix of text only data (TS and PreTS) do not follow the response format explicity specified and therefore it is unlikely the response would be correctly counted towards their score. This is a great example showing that we have instilled knowledge and reasoning abilities over time-series but the instruction following capabilities have deteriorated compared to the models trained on both text and the TS Instruct dataset. Note that this lapse only applies for reasoning on multi-modal data and that the performance on purely text benchmarks has not deteriorated.

## F.2    TS INSTRUCT

Here we show some examples from our TS Instruct benchmark which was generated for the quantitative analysis task. The first example is shown below in Table 14. Here we show the a failure case of current time-series analysis methodologies. Because these models rely on normalization as part of the tokenization process, they fail to apply their knowledge to real valued time-series. The models are fed a encoded version of the time-series which after normalization loses its original scale. Even when the model correctly produces the steps used, the final answer is incorrect since it has limited understanding of the original values in the time-series. We also see that the LLama 3.1-8B model tends to create longer and more elaborate answers but that do not necessarily answer the question (for example computing the absolute rate of change instead of the slope).

In our next example in Table 15 we show a case study from the decision making tasks in the TS Instruct quantitative analysis. While all of the models correctly identify that the reservoir levels are rising and it may be a good idea to increase the extraction rate with caution, the LLama 3.1-8B SOTA model provides a longer output with several halucinations and incorrect statements. For example it assumes that each data point corresponds to months, however that is not neccissarily true, just that the data spans several months (the x value could be days or weeks). It also provides a good example of the normalization problem wherein the trend under **Recent increase:** is growing but the values are not correct. It assumes supplementary information like the reservoir level is not critically low and that the increasing reservoir levels mean that the extraction rate is not sufficient (this data was provided by the user and is not directly tied to the water levels).

**Prompt:** The dataset represents fluctuations in a certain variable over time, likely affected by external factors. Based on this, what could be the most likely interpretation of the trend observed in the time-series data?
A: A steady increase indicates a long-term positive external influence.
B: Frequent large spikes suggest random noise without any underlying trend.
C: A systematic drop followed by recovery indicates a seasonal effect.
D: Significant fluctuations followed by a general downward trend suggest an underlying systemic issue.

**Ground Truth Answer:** The correct answer is D: Significant fluctuations followed by a general downward trend suggest an underlying systemic issue.

*(Optional Explanation):* The observed significant fluctuations along with the overall downward trend indicate potential instability or decline in the underlying variable, pointing to systemic issues rather than simply seasonal effects or random noise.

**PreOrcaTS Model Answer:** Based on the time-series data provided, the most likely interpretation of the trend observed is: D: Significant fluctuations followed by a general downward trend suggest an underlying systemic issue.

This interpretation is supported by the presence of multiple peaks and troughs in the data, indicating variability that could be influenced by external factors. The overall downward trend suggests that despite these fluctuations, there is a persistent decline in the variable being measured, which could indicate a systemic issue that needs to be addressed.

**OrcaTS Model Answer:** Based on the time-series data provided, the most likely interpretation of the trend observed is: D: Significant fluctuations followed by a general downward trend suggest an underlying systemic issue.

This interpretation is supported by the presence of multiple peaks and troughs, indicating variability in the data, followed by a gradual decline. This pattern could suggest that external factors are influencing the variable, leading to a systemic issue that needs to be addressed.

**TS Model Answer:** Based on the analysis of the time-series data, I classify the trend as representing: A steady increase indicates a long-term positive external influence.

This suggests that the variable is consistently rising, possibly due to factors like increased demand or improved conditions.

**PreTS Model Answer:** Based on the time-series data, I classify the trend as representing: A steady increase indicates a long-term positive external influence.

This could be due to consistent growth in demand or an improvement in the underlying process being measured. The gradual rise suggests stability and potential for further growth, which is a positive sign for the variable in question.

**LLama 3.1-8B Model Answer:** A: A steady increase indicates a long-term positive external influence.

The given time-series data shows a general upward trend over time, with some fluctuations. The data points are mostly increasing, and there are no significant drops or spikes that would indicate a seasonal or random effect. This suggests that there is a long-term positive external influence at play, which is driving the steady increase in the variable being measured. This could be due to various factors such as a growing market, increasing demand, or a positive economic trend.

Table 13: Example failure case for multi-modal LLMs in interpreting time-series trends. Variability in model interpretations demonstrates potential weaknesses in trend analysis and the risk of hallucinating conclusions from the same dataset.

**Prompt:** Compute the rate of change between the first and last values in the dataset.

**Ground Truth Answer:** Of course! To compute the rate of change, we'll use the formula: Rate of Change $= \frac{\Delta y}{\Delta x} = \frac{y_{\text{last}} - y_{\text{first}}}{x_{\text{last}} - x_{\text{first}}}$. From the data, let's assume these values:

- First value at $x = 0$ (let's say this is 202000)

- Last value at $x = 112$ (let's say this is 199000)

Now, substituting these values into the formula: Rate of Change $= \frac{199000 - 202000}{112 - 0} = \frac{-3000}{112} \approx -26.79$. Thus, the rate of change is approximately **-26.79 per time unit**, indicating a decrease in the data values over the observed period.

Time Series Sample 45

**PreOrcaTS Model Answer:** Sure! To compute the rate of change between the first and last values in the dataset, we can use the formula: Rate of Change $= \frac{y_{\text{last}} - y_{\text{first}}}{x_{\text{last}} - x_{\text{first}}}$. From the time-series data you provided, let's assume the following values:

- $y_{\text{first}} = 100$ (value at the first time point)

- $y_{\text{last}} = 200$ (value at the last time point)

- $x_{\text{first}} = 0$ (first time point)

- $x_{\text{last}} = 44$ (last time point)

Now, substituting these values into the formula: Rate of Change $= \frac{200 - 100}{44 - 0} = \frac{100}{44} \approx 2.27$. This means that the rate of change in the dataset is approximately **2.27 units per time point**. This indicates a steady increase in the values over the observed period.

**LLama 3.1-8B Model Answer:** To compute the rate of change between the first and last values in the dataset, we need to follow these steps:

1. Identify the first and last values in the dataset.

2. Calculate the difference between the first and last values.

3. Divide the difference by the number of time periods (or the number of data points) to get the average rate of change.

Let's identify the first and last values in the dataset:

- The first value is: 1.052

- The last value is: 0.2404

Now, let's calculate the difference between the first and last values: Difference $= 0.2404 - 1.052 = -0.8116$. Since we are calculating the rate of change, we will use the absolute value of the difference: $|\text{Difference}| = 0.8116$. Now, calculate the average rate of change over 101 data points: Rate of Change $= \frac{0.8116}{100} = 0.008116$. So, the average rate of change between the first and last values is approximately **0.008116**. When expressed as a percentage: Percentage Rate of Change $= 0.008116 \times 100 = 0.81\%$. Therefore, the rate of change between the first and last values is approximately **0.81% per time period**.

Table 14: Example failure case for multi-modal LLM's in mathematical reasoning. Computing values over the real-life values becomes difficult since the models rely on normalization of the time-series before tokenization. While the process may be generally correct, the models will often make up or hallucinate values causing the actual answers to be inaccurate.

**Prompt:** Hi there! I'm managing a water supply project, and I've been tracking our reservoir levels over the past few months. Currently, I'm debating whether to increase our water extraction rate now to meet growing demand or to maintain our current extraction rate to ensure sustainability. Considering the recent water levels, what do you recommend?

**Ground Truth Answer:** After analyzing the reservoir data, it's evident that the water levels have shown a consistent upward trend over the last several weeks, reaching levels higher than we've seen in recent months. This suggests that there's ample supply available. However, while the increase looks promising, I recommend that you proceed with caution and slightly increase the extraction rate, rather than a full-scale jump. This way, you can meet the immediate demand without risking long-term sustainability. Balancing rapid extraction with conservation is key here!

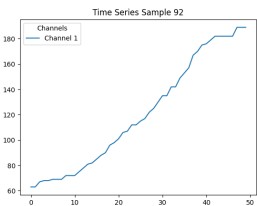

**PreOrcaTS Model Answer:** Based on the time-series data of your reservoir levels, I've observed a significant upward trend over the past few months, indicating a steady increase in water availability. However, there are some fluctuations in the data, suggesting that while the overall trend is positive, there may be periods of volatility. Given the current demand and the rising levels, I recommend increasing your water extraction rate now. This proactive approach will help meet the growing demand while also ensuring that you're making the most of the available resources. However, it's essential to monitor the levels closely to adjust your extraction rate if the trend shifts or if any unexpected changes occur.

**OrcaTS Model Answer:** Based on the time-series data you've provided, I've observed a significant upward trend in reservoir levels over the past few months, particularly in the last few weeks. This indicates that the current extraction rate is not causing a depletion of the reservoir, and there's ample water available to meet the growing demand. Therefore, I recommend increasing your water extraction rate now to capitalize on the current surplus and ensure that you can meet the anticipated demand without compromising sustainability in the future.

**PreTS Model Answer:** Looking at the time-series data, there's a noticeable upward trend in the reservoir levels over the past few weeks, indicating a recovery from previous lows. However, there are also some fluctuations that suggest potential volatility. Given this context, I recommend increasing the extraction rate now. The rising levels indicate that the reservoir is replenishing, and meeting the growing demand could help manage the supply effectively before any potential downturns occur. It's a strategic move to balance current needs with sustainability.

**LLama 3.1-8B Model Answer:** To provide a recommendation, I'll analyze the given time-series data representing the reservoir levels over the past few months. The data is presented as a sequence of numbers, where each number represents the reservoir level at a specific point in time.
Upon examining the data, I notice the following trends:

- **Increasing trend**: The reservoir levels have been increasing over time, with some fluctuations. This suggests that the current extraction rate is not depleting the reservoir at an alarming rate.

- **Seasonal fluctuations**: There are noticeable seasonal fluctuations in the reservoir levels, with higher levels during the summer months (e.g., July, August) and lower levels during the winter months (e.g., January, February).

- **Recent increase**: The reservoir levels have been increasing rapidly over the past few months, with a significant jump in the last few data points (e.g., 9.7, 9.7, 9.7, 9.7, 9.7, 10.19, 10.19, 10.19).

Considering the increasing trend and recent increase in reservoir levels, I recommend **increasing the water extraction rate** to meet the growing demand. This decision is based on the following reasoning:

- **Current sustainability**: The reservoir levels are not critically low, and the current extraction rate is not depleting the reservoir at an alarming rate.

- **Growing demand**: The increasing trend in reservoir levels suggests that the current extraction rate is not sufficient to meet the growing demand.

- **Opportunistic increase**: With the recent increase in reservoir levels, it is an opportune time to increase the extraction rate to meet the growing demand without compromising the sustainability of the reservoir.

To ensure the sustainability of the reservoir, it is essential to monitor the reservoir levels closely and adjust the extraction rate as needed. This may involve:

- Regular monitoring of reservoir levels to ensure they remain within a sustainable range.

- Adjusting extraction rates based on the reservoir levels, accounting for growing demand and seasonal fluctuations.

- Implementing conservation measures, such as reducing non-essential water usage or implementing water-saving technologies, to reduce the demand on the reservoir.

Table 15: Example of model outputs for reservoir level analysis. Models provide varying recommendations based on the same time-series data, highlighting their reasoning and suggestions for water extraction rate management.

