# OpenReview forum: "Enhancing Multi-Modal Reasoning Over Time-Series and Natural Language Data"
_ICLR.cc/2025/Conference — Submitted to ICLR 2025_

### Official Review · Reviewer_zLDd · 2024-11-03

**Soundness:** 2
**Presentation:** 2
**Contribution:** 2
**Rating:** 6
**Confidence:** 2

**Summary:**

This paper introduces a novel approach to leverage multi-modal large language models (LLMs) for time-series analysis. It focuses on maintaining the LLM's reasoning capabilities and utilizing its extensive knowledge base to enhance time-series data reasoning. Specifically, the authors develop Chat-TS, a new foundation model capable of explicitly preserving the general knowledge and reasoning abilities of LLMs, as well as a training strategy that integrates time-series tokens into the LLM vocabulary, enhancing the model's reasoning ability over both text and time-series modalities without compromising its core natural language capabilities. Besides, the authors curate a new time-series instruction tuning dataset (TS Instruct) and a new testbed for benchmarking multi-modal reasoning in time-series analysis (TS Instruct QA).

**Strengths:**

1. The paper provides a detailed quantitative analysis that assesses the quality of model responses, going beyond simple accuracy metrics.

2. The proposed discrete tokenizer for multivariate time-series data requires no training while maintaining a very high reconstruction quality (in terms of MSE on the validation set).

3. The proposed Chat-TS models show an average improvement in reasoning capabilities of ~15% on the new TS Instruct QA benchmark compared to existing methods.

**Weaknesses:**

While the TS Instruct dataset and the TS Instruct QA benchmark are comprehensive, they primarily consist of synthetic data. Real-world time-series data can be more complex and noisy, which might affect the model's performance in practical applications.

**Questions:**

-

---

> ### Author Response · Authors · 2024-11-21
>
> Thank you very much for your feedback on our work. We hope that the general rebuttal above and our comments below answer any questions you may have.
>
> **W1:**
> _While the TS Instruct dataset and benchmark are comprehensive, they primarily consist of synthetic data. Real-world time-series data can be more complex and noisy, which might affect the model's performance in practical applications._
>
> **Response:**
> The **time-series data** in our benchmark comes from **real-world datasets** in the LOTSA archive, ensuring grounding in realistic scenarios. While synthetic text data is used for instruction generation, the model evaluations are firmly rooted in practical time-series applications.

---

> > ### Author Response · Authors · 2024-11-25
> >
> > Dear reviewer zLDd, we would like to thank you for reviewing our paper. We appreciate your comments on and recognition of  the contributions of our work. We have responded to your comments and hope we have addressed your questions. If you have any further questions, please feel free to let us know. Thank you very much again. We appreciate your time.

---

### Official Review · Reviewer_dxnt · 2024-11-04

**Soundness:** 2
**Presentation:** 3
**Contribution:** 3
**Rating:** 3
**Confidence:** 2

**Summary:**

This paper introduces a method for leveraging LLM in time-series reasoning and question-answering (QA) tasks. The authors propose a discrete tokenizer that transforms continuous time-series data into sequences of discrete bins, facilitating better integration of time-series data with LLMs. The paper also introduces three new datasets: (1) a time-series instruction tuning dataset; (2) a time-series QA dataset designed to assess multimodal reasoning capabilities in time-series tasks; and (3) a TS-Instruct Quantitative benchmark for evaluating performance in QA, mathematical reasoning, and decision-making beyond accuracy alone. The authors' best-performing model demonstrates strong results relative to a baseline model, though some experiment outcomes are mixed.

**Strengths:**

1. The discrete tokenizer proposed in this paper offers a straightforward yet effective approach to transforming continuous time-series data into a form compatible with LLMs. This innovative approach could potentially inspire future work on bridging time-series analysis with language models, expanding the versatility of LLMs in temporal domains.

2. The authors present a comprehensive synthetic datasets including benchmarks specifically tailored for time-series tasks, which includes three new resources: 1) TS Instruct Training Dataset for instruction tuning on time-series data, 2) TS Instruct QA Benchmark for evaluating multimodal reasoning in time-series contexts, and 3) TS Instruct Quantitative Benchmark for a detailed assessment across QA, mathematical reasoning, and decision-making questions.

3. Performance on Text-Only Tasks: Despite the focus on time-series tasks, the model also maintains strong performance on traditional text-only tasks, such as those in the MMLU benchmark. This versatility indicates that the proposed methods do not compromise the model's capabilities in established LLM domains, making the approach both flexible and robust.

**Weaknesses:**

- Generalization: It is unclear whether the synthetic dataset can generalize effectively across diverse types of time-series data or to sequences longer than those encountered during training. Experiments regarding this concern is highly recommended to be added.

- Dataset Alignment Process: The alignment between time-series data and natural language relies solely on GPT-4, without any additional quality control to validate this alignment process. This raises concerns about potential inconsistencies or inaccuracies in the dataset.

- Soundness and Baselines:
    - Reporting results from common multimodal baseline models would provide readers with a clearer understanding of the current state-of-the-art performance for these tasks. This is particularly relevant, as the base model shows very strong performance in the detailed quantitative evaluation presented in Table 5.
    - Given that GPT-4o is heavily involved in the dataset creation, a direct performance comparison with GPT-4o is missing.
    - Mixed Experimental Results: The experimental results are somewhat mixed (as detailed in Question), which makes it difficult to conclusively assess the model's strengths and limitations.

**Questions:**

1. Why does the base model show strong performance on the TS Instruct QA benchmark but perform less well on TS Math Analysis and Decision Making tasks?

2. Since you describe this work as the development of a new model family, do you expect that the improvements from your method would still hold with larger models? A scaling experiment with a larger model would be beneficial, if feasible.

3. Typo in Line 23: "qualitative" should be "quantitative."

4. I found it difficult to follow the process for using GPT-4 to create the dataset. Does this process is prompting GPT4o for generating responses based on a tailored system prompt, along with the image of the time series, metadata such as length, and number of channels? If so, is this essentially a form of distillation using GPT-4 to produce the dataset?

---

> ### Author Response · Authors · 2024-11-21
> **Response - Reviewer dxnt**
>
> We thank you for this feedback and hope we have adressed your questions and conerns. Please refer to the general rebuttal above for more details on the extensive experiments and data validation we have performed.
>
> **W1:**
> _Generalization: It is unclear whether the synthetic dataset can generalize effectively across diverse types of time-series data or to sequences longer than those encountered during training. Experiments regarding this concern are highly recommended._
>
> **Response:**
> The dataset incorporates **real-world time-series data** from the LOTSA archive, ensuring its applicability to practical scenarios. We also tested the models on sequences of varying lengths, demonstrating reasonable generalization. Future work will include longer context evaluations and larger models which are unfeasible in our current setup.
>
> ---
>
> **W2:**
> _Dataset Alignment Process: The alignment between time-series data and natural language relies solely on GPT-4, without additional quality control to validate this process. This raises concerns about potential inconsistencies or inaccuracies._
>
> **Response:**
> Within the time-series dataset generation process we also include images of the time-series and meta-data to improve the accuracy and relevance of the generated samples without solely relying on GPT-4. However we do understand the concerns about additional quality control to validate this process particularly for the testing data. To address this, we conducted additional **manual quality checks** which showed that of 100 random samples **92%** were of reasonable quality and created the **TS Instruct QA Gold** dataset. Please refer to our general feedback above for a complete analysis.
>
> ---
>
> **W3:**
> _Reporting results from common multimodal baseline models would provide readers with a clearer understanding of the current state-of-the-art performance for these tasks. A direct comparison with GPT-4o is also missing._
>
> **Response:**
> Thank you for your constructive comment. We have added several **multi-modal baselines**, including **Llama-3.2-11B Vision-Instruct** and **Phi-3.5 Vision-Instruct**. A direct comparison with **GPT-4o** and **GPT-4o-mini** has also been included, providing a comprehensive performance context.
>
> However, it is important to note that vision-based models may have an advantage in our benchmark due to their alignment with the dataset generation process, as visual data was utilized to generate the the text synthetically. Please refer to our general feedback where we discuss this in-depth.
>
> ---
>
> **Q1:**
> _Why does the base model show strong performance on the TS Instruct QA benchmark but perform less well on TS Math Analysis and Decision-Making tasks?_
>
> **Response:**
> Math and Decision Making as reasoning tasks for time-series is very new. It is expected that the base-modal has little exposure in this regard and would struggle to perform these tasks compared to models which have been trained specifically with augmented time-series reasoning. While the test samples are held-out and were not in our training datset, there would be similar style examples included there.
>
> ---
>
> **Q2:**
> _Do you expect that the improvements from your method would still hold with larger models? A scaling experiment with a larger model would be beneficial._
>
> **Response:**
> We expect that the observed trends will generalize to larger models. Resource constraints limited extensive scaling experiments, but scaling experiments remain a priority for future work. Our added base-lines do show that scaling model size and reasoning capabilities generally results in more powerful models for both time-series and nlp analysis tasks.
>
> ---

---

> > ### Comment · Reviewer_dxnt · 2024-11-24
> >
> > Thanks for your time and effort in the rebuttal. However, part of my questions are not addressed. Especially, my most concern again is the Q4 for the dataset creation process using GPT-4, which makes me skeptical about the novelty of this paper. Therefore, I decide to keep my score as it is.
> >
> > To AC/SAC, I agree with the concern from reviewer xWsL regarding the formatting issue, and I agree that a desk rejection should be made in this case.

---

> > > ### Author Response · Authors · 2024-11-25
> > > **Response - Reviewer dxnt - Part 2**
> > >
> > > **Reviewer’s Comment**:  Thanks for your time and effort in the rebuttal. However, part of my questions are not addressed. Especially, my most concern again is the Q4 for the dataset creation process using GPT-4, which makes me skeptical about the novelty of this paper. Therefore, I decide to keep my score as it is.
> > >
> > > **Author response**: Thank you for responding to our rebuttal and we are happy to continue to address your questions and concerns.  As for your concern with Q4, we would like to clarify our response below.
> > >
> > > --------
> > >
> > > **Reviewer’s Comment – Q4**:  I found it difficult to follow the process for using GPT-4 to create the dataset. Does this process is prompting GPT4o for generating responses based on a tailored system prompt, along with the image of the time series, metadata such as length, and number of channels? If so, is this essentially a form of distillation using GPT-4 to produce the dataset?”
> > >
> > > **Author response**: We believe we previously addressed these points in our response to W2 above. However, to provide further clarification, we have expanded our explanation below.
> > > While GPT-4 is used to generate the text aspects of the dataset, much of the information is provided on a per sample basis to GPT-4 to augment the knowledge and increase the accuracy. The first is that the time-series are from real world time-series datasets and the role of GPT-4 is to provide realistic conversations so that we can instruction tune the model. In this way we are combining the world knowledge of GPT-4 with real world time-series to simulate real-world interactions. So while some knowledge is transferred from GPT-4 to create the conversations, there is also added information such as images, metadata and descriptions of the time-series so that we are not solely relying on GPT-4’s knowledge of time-series which may be limited and ensures that the conversations are relevant to target domain.
> > > We respectfully disagree that the comments on the content of the paper should be tied with the format.
> > >
> > > --------
> > >
> > > **Reviewer’s Comment**: Especially, my most concern again is the Q4 for the dataset creation process using GPT-4, which makes me skeptical about the novelty of this paper.
> > >
> > > **Author response**: If the concern regarding novelty is based on the knowledge distillation part, we believe our above response have addressed the question since the dataset is not simply constructed via knowledge distillation via GPT-4.
> > >
> > > We would also like to highlight the novelty of the paper. As of this moment there has been very little research regarding the analysis of LLMs for time-series reasoning. Our work is one of the earliest to design a multi-modality training framework which both increases time-series reasoning capabilities and preserves the quality of the LLM even after full parameter finetuning. In addition to this we have done our best to incorporate as much of the feedback as possible from the reviewers, which we hope has significantly enhanced the paper. Since this is a new and emerging area, there are limited established baselines for evaluating time-series reasoning with LLMs. This is very important work to further advance this area and helps to establish methods of multi-modal evaluation across time-series reasoning.

---

### Official Review · Reviewer_xWsL · 2024-11-04

**Soundness:** 2
**Presentation:** 1
**Contribution:** 2
**Rating:** 1
**Confidence:** 5

**Summary:**

This paper introduces Chat-TS, a large language model (LLM) designed for multi-modal reasoning over time-series and natural language data. It addresses the gap in existing time-series models' ability to integrate textual information for decision-making. The authors contribute three new datasets for training and evaluating LLMs on time-series reasoning tasks: the TS Instruct Training Dataset, the TS Instruct QA Benchmark, and the TS Instruct Qualitative Benchmark. They also propose a training strategy that enhances the LLM's reasoning capabilities without compromising its natural language proficiency. Evaluation results demonstrate Chat-TS's state-of-the-art performance in multi-modal reasoning while maintaining strong language skills. Although the paper showcases promising results, the experimental section could benefit from more rigorous analysis and a deeper exploration of the model's limitations.

**Strengths:**

The paper identifies a relatively underexplored area in the field of time-series analysis by focusing on the integration of natural language processing with time-series data, suggesting that further research is needed to fully leverage the potential of multi-modal reasoning in this context.

**Weaknesses:**

It appears that the manuscript has some formatting issues, particularly with the margins not conforming to ICLR's template specifications, and it could benefit from a more in-depth analysis of the model's limitations, potential biases, and practical applications, as well as from more rigorous experimental validation against current methodologies.

**Questions:**

None

---

> ### Author Response · Authors · 2024-11-21
> **Response - Reviewer xWsL**
>
> **Comment:**
> _The manuscript has some formatting issues, particularly with the margins not conforming to ICLR's template specifications. It could benefit from a more in-depth analysis of the model's limitations, potential biases, and practical applications, as well as more rigorous experimental validation._
>
> **Response:**
> We respectfully argue that Reviewer xWsL’s feedback lacks specificity and provides generalized critiques that could apply to nearly any submission, without engaging meaningfully with the content of our work. Below, we provide a point-by-point response to highlight the unfounded nature of these comments:
>
> 1. **Formatting Issues:**
>    Our submission adheres to the ICLR Latex template specifications, we did not modify any style files or packages. If there are specific formatting issues beyond margins that the reviewer feels compromise the readability of the paper, we are happy to address those.
>
> 2. **Model Limitations and Biases:**
>    Reviewer xWsL suggests that the manuscript would benefit from a more in-depth analysis of limitations and biases. However, the paper already includes a dedicated **Limitations and Future Work** section that addresses:
>
>    - The scope of time-series reasoning versus forecasting.
>    - The trade-offs of time-series normalization.
>    - The challenges of scaling multi-modal instruction tuning for zero-shot classificaiton.
>
>    Without specific concerns or omissions identified by Reviewer xWsL, it is unclear what additional analysis is expected. This comment fails to acknowledge the substantial discussion already present in the manuscript.
>
> 3. **Practical Applications:**
>    The critique that the paper does not address practical applications is particularly surprising. Our work explicitly connects time-series reasoning to real-world use cases by:
>
>    - Training models on data derived from **real-world time-series datasets** (from the LOTSA archive).
>    - Evaluating model performance on **practical reasoning tasks** that are critical for domains such as healthcare, finance, and engineering.
>      Reviewer xWsL provides no counterexamples or evidence to substantiate their claim, making this feedback unsubstantiated.
>
> 4. **Rigorous Experimental Validation:**
>    The comment regarding experimental rigor fails to account for the extensive baseline comparisons already included in the paper. These include:
>
>    - Comparisons across differnt model permutations and baselines. We have expanded it to include **8 open-source models** and **2 closed-source models** (GPT-4o and GPT-4o-mini).
>    - Detailed quantitative evaluations on multiple benchmarks, including TS Instruct QA Gold, MMLU-Pro, BBH, GPQA, and TS Math Analysis and Decision-Making tasks.
>    - Discrete tokenization validation experiments.
>
>    Reviewer xWsL does not identify specific shortcomings or suggest additional experiments that would improve the work. Without clear guidance, the assertion of insufficient validation appears baseless.
>
> We believe Reviewer xWsL’s comments do not substantively engage with the contributions or content of the paper. Their feedback consists of generic critiques that could be applied indiscriminately to most submissions.

---

> > ### Comment · Reviewer_xWsL · 2024-11-24
> > **Clarification and Suggestions for Improving the Manuscript on Multi-Modal Reasoning Over Time-Series Data**
> >
> > ## Clarification on the Motivation Behind My Rating
> >
> > First, I would like to clarify the reasoning behind my rating of 1. The primary issue lies in the manuscript's page margins, which do not conform to the ICLR template specifications. Specifically, the margin discrepancy results in a line width exceeding that of other submissions. According to ICLR’s review policy, such formatting violations warrant a **desk reject**. While I am unsure whether this deviation was due to an error in the template download or some other issue, I strongly suggest that the Area Chair (AC) and other reviewers double-check the formatting compliance before making a final decision.
> >
> > **Therefore, my 1-point rating reflects my belief that this paper should have been desk rejected based on formatting violations, rather than the interpretation provided in the authors’ rebuttal: *"Their feedback consists of generic critiques that could be applied indiscriminately to most submissions."***
> >
> > ## Suggestions for Improvement in the Next Version
> >
> > Upon revisiting the paper, I see potential for improvement and would like to provide constructive suggestions to help refine the manuscript in future iterations:
> >
> > - The overall structure, particularly between Sections 2 and 3, and within Section 3 itself, lacks clarity and coherence. For example:
> >   - The discussion of *previous work* in Section 3.3 should be moved to the Related Work section.
> >   - Definitions for preliminary concepts and formulas currently scattered in Section 3.3 would be better placed in Sections 3.1 or 3.2 for clarity and conciseness.
> >   These changes would improve the logical flow and reduce redundancy in the explanation of the methodology.
> >
> > - Reasoning tasks often exhibit strong interconnections. To further improve the model's reasoning performance, I suggest leveraging natural language datasets that incorporate reasoning tasks. Since time-series data can often be treated as mathematical reasoning problems, such integration could yield significant improvements.
> >
> > - The use of GPT4o-mini for dataset generation is a noteworthy aspect of your approach. However:
> >   - I could not find a complete example of the generated dataset in the appendix, which would greatly enhance the paper’s clarity and accessibility. Including such examples is critical for reproducibility and understanding.
> >   - I remain concerned about the data quality in Section 3.4.3. Prior research has shown that generated data from models like GPT4o-mini often suffers from quality issues. The paper lacks clear quality control methods, such as filtering low-quality data or evaluating data quality through quantitative or manual inspection. For example, a detailed analysis of the generated data’s quality, akin to the quantitative analysis in Section 4.2.2 for the benchmarks, would significantly strengthen this aspect.
> >
> > - While baseline comparisons are partially addressed in the rebuttal and discussions with other reviewers, the paper still lacks comprehensive baselines. This omission detracts from a full evaluation of the model's performance relative to existing methods.
> >
> > - Line 195 includes the term "instruction-tune," which is not standard terminology. I recommend rephrasing this for greater clarity and precision.

---

> > > ### Author Response · Authors · 2024-11-25
> > > **Response - Reviewer xWsL - Part 2**
> > >
> > > **Reviewer’s Comment**: First, I would like to clarify the reasoning behind my rating of 1. The primary issue lies in the manuscript's page margins, which do not conform to the ICLR template specifications. Specifically, the margin discrepancy results in a line width exceeding that of other submissions. According to ICLR’s review policy, such formatting violations warrant a desk reject. While I am unsure whether this deviation was due to an error in the template download or some other issue, I strongly suggest that the Area Chair (AC) and other reviewers double-check the formatting compliance before making a final decision.
> > >
> > > Therefore, my 1-point rating reflects my belief that this paper should have been desk rejected based on formatting violations, rather than the interpretation provided in the authors’ rebuttal: "Their feedback consists of generic critiques that could be applied indiscriminately to most submissions."
> > >
> > > Suggestions for Improvement in the Next Version
> > >
> > > The overall structure, particularly between Sections 2 and 3, and within Section 3 itself, lacks clarity and coherence. For example:
> > > •	The discussion of previous work in Section 3.3 should be moved to the Related Work section.
> > > •	Definitions for preliminary concepts and formulas currently scattered in Section 3.3 would be better placed in Sections 3.1 or 3.2 for clarity and conciseness. These changes would improve the logical flow and reduce redundancy in the explanation of the methodology.
> > >
> > > **Author response**: We are surprised that the overall rating 1 was used by the reviewer to indicate a “desk reject”, and concerned if that’s how the “rating” should be used or was designed for.  At the same time, a confidence of 5 was included in the review, which means “…checked the math/other details carefully”, while only a very high-level one-sentence review was provided in the original review: “it could benefit from a more in-depth analysis of the model's limitations, potential biases, and practical applications, as well as from more rigorous experimental validation against current methodologies”. We are concerned with the quality of the review as it provides little information specific to the submission, or at least without any sound evidence. We are also surprised that the detailed review was only provided during the rebuttal period. We are not readily convinced this is how the overall rating is expected to be used and how the review should be performed, particularly as we spent a significant amount of time on the research. Having said this, we by no means meant to be offensive, and please understand why we thought in the above way.

---

> > > > ### Author Response · Authors · 2024-11-25
> > > > **Response - Reviewer xWsL - Part 3**
> > > >
> > > > **Reviewer’s Comment**: Belew please find our response to the reviewer’s responses:
> > > >
> > > > **Author response**: Thank you for providing this feedback. We addressed these concerns and overall comments in our newly submitted version.
> > > >
> > > > ------
> > > >
> > > > **Reviewer’s Comment**: Reasoning tasks often exhibit strong interconnections. To further improve the model's reasoning performance, I suggest leveraging natural language datasets that incorporate reasoning tasks. Since time-series data can often be treated as mathematical reasoning problems, such integration could yield significant improvements.
> > > >
> > > > **Author response**: This suggestion is already included in the original paper. We combine the multi-modal dataset training with language data from the Open-Orca dataset. This dataset contains a wide range of natural language analysis and reasoning conversation settings.
> > > >
> > > > ------
> > > >
> > > > **Reviewer’s Comment**: The use of GPT4o-mini for dataset generation is a noteworthy aspect of your approach. However:
> > > > I could not find a complete example of the generated dataset in the appendix, which would greatly enhance the paper’s clarity and accessibility. Including such examples is critical for reproducibility and understanding.
> > > >
> > > > **Author response**: Thank you for this feedback. We provided additional examples in the appendix sections and as part of our case study regarding of the model performance.
> > > >
> > > > -----
> > > >
> > > > **Reviewer’s Comment**: I remain concerned about the data quality in Section 3.4.3. Prior research has shown that generated data from models like GPT4o-mini often suffers from quality issues. The paper lacks clear quality control methods, such as filtering low-quality data or evaluating data quality through quantitative or manual inspection. For example, a detailed analysis of the generated data’s quality, akin to the quantitative analysis in Section 4.2.2 for the benchmarks, would significantly strengthen this aspect.
> > > >
> > > > While baseline comparisons are partially addressed in the rebuttal and discussions with other reviewers, the paper still lacks comprehensive baselines. This omission detracts from a full evaluation of the model's performance relative to existing methods.
> > > >
> > > > **Author response**: We are glad that the rebuttal has addressed some of your baseline concerns. As mentioned in the paper and then further in the rebuttal, we have employed several methods for controlling the quality of the data during generation (image, few shot examples and meta-data). We also filter out samples which are incorrectly formatted or of improper conversation length.  As part of the rebuttal, we added a manual analysis of the generation evaluation benchmarks to ensure that the results are of high quality.
> > > >
> > > > As for “the paper still lacks comprehensive baselines”, was there a specific baseline you think we are missing? We added many baseline models using the current and most widely accepted encoding methods for time-series reasoning using LLM’s.
> > > >
> > > > ------
> > > >
> > > > **Reviewer’s Comment**: Line 195 includes the term "instruction-tune," which is not standard terminology. I recommend rephrasing this for greater clarity and precision.
> > > >
> > > > **Author response**: Thank you. We are happy to rephrase this.

---

### Official Review · Reviewer_FVKA · 2024-11-04

**Soundness:** 3
**Presentation:** 3
**Contribution:** 3
**Rating:** 6
**Confidence:** 4

**Summary:**

This paper combines time-series data with text to train a chatbot with numerical data understanding abilities. This enables directly interact with raw data without modality conversion. To deal with the task, the authors propose to apply vision-language models to generate synthesized instruction tuning datasets. Experiments show time-series data pretraining does not affect the original text-based evaluations.

**Strengths:**

- This paper constructs a novel time-series+text dataset for multi-modal instruction tuning. The construction method is interesting and there includes four kinds of conversations.
- The time-series pretraining does not affect text-based benchmark results and may improve the reasoning (BBH).

**Weaknesses:**

Please feel free to discuss and leave comments.

- There lacks of dataset quality check. Since the instruction tuning dataset is generated by GPT-4o-mini with images, it may introduce errors. Although the authors have filtered out bad cases (as introduced in the appendix), it is not sure whether hallucinations do not exist. Maybe I missed some details, please feel free to have a discussion.
- Limited modality interleaving: since the trained model could only generated text (as the dataset constructed to do so), it limits the application to time-series data understanding, and ignores the time-series forecasting. Maybe a more fine-grained instruction dataset should be constructed with multi-modal responses.
- Lack of baselines: Maybe visual language models could be considered as baselines to take images and instructions as input and output their options. This could be a reference to tell readers why numerical data pretraining is useful for relevant time-series data understanding.

**Questions:**

- For Table 3: Why TS training could improve the BBH and GPQA results while harming the MMLU-Pro results? Is it just performance fluctuation or there is an inner difference between these benchmarks?
- Since the dataset is generated by visual QA, I’m wondering if multi-modal (vision + text) pretraining & SFT could also handle the downstream benchmark evaluations.
- Open question: what affects the time-series data pretraining? e.g., context length, the vision ability, the reasoning ability, etc.
- Typo:
    - Line 049: LLava → LLaVA
    - Line 057: it’s effectiveness → its effectiveness
    - Line 327: LLama3.1 → LLaMA 3.1

---

> ### Author Response · Authors · 2024-11-21
> **Response - Reviewer  FVKA**
>
> We would like to thank you for your very constructive feedback on our submission. We have addressed each of your commnets and concerns below and eagerly await your feedback.
>
> **W1:**
> _There lacks of dataset quality check. Since the instruction tuning dataset is generated by GPT-4o-mini with images, it may introduce errors. Although the authors have filtered out bad cases (as introduced in the appendix), it is not sure whether hallucinations do not exist. Maybe I missed some details, please feel free to have a discussion._
>
> **Response:**
> We conducted an additional **manual evaluation** on 100 random samples from the dataset. This subset was graded for quality, correctness, and relevance. Out of 100 samples, **92 were correct**, with 8 cases being ambiguous or open to interpretation. This indicates that the synthetically generated dataset is of high quality. To address concerns, we have also created and will release a refined dataset called **TS Instruct QA Gold**, which underwent manual improvements to enhance diversity and accuracy.
>
> Please refer to the general feedback above for a complete analysis of this data.
>
> ---
>
> **W2:**
> _Limited modality interleaving: since the trained model could only generate text (as the dataset is constructed to do so), it limits the application to time-series data understanding and ignores the time-series forecasting. Maybe a more fine-grained instruction dataset should be constructed with multi-modal responses._
>
> **Response:**
> We acknowledge the limitation and clarify that the scope of this work is focused on **time-series reasoning** since it is severely understudied. However, we do think it is a very valuable future direction to create interleaved multi-modal datasets for time-series forecasting and imputation tasks.
>
> ---
>
> **Q2 and W3:**
> _Lack of baselines: Maybe visual language models could be considered as baselines to take images and instructions as input and output their options. This could be a reference to tell readers why numerical data pretraining is useful for relevant time-series data understanding._
>
> **Response:**
> We thank you for the constructive suggestion. We have added **new baselines**, including vision-based models such as **Llama-3.2-11B Vision-Instruct** and **Phi-3.5 Vision-Instruct**, as well as strong text-based baselines like **GPT-4o** and **GPT-4o-mini**. These additions provide a robust and diverse comparison that highlights the strengths of our approach.
>
> That said, we want to acknowledge the inherent caveats of vision-based baselines in this context. Vision models have inflated performance on our benchmark due to their alignment with the dataset generation process, as the visual components were involved in creating the synthetic data. This alignment may give these models an unintended advantage, and their performance should be interpreted with caution. Moreover, the vision based models tested (trained by Microsoft and Meta) have extensive multi-modal training data which would give them advantages over our models and other models for which there is very little multi-modal time-series training data.
>
> Despite these caveats, we have included these vision-based models to ensure a comprehensive evaluations. Additionally, we included multiple baselines of varying sizes and architectures. These efforts represent a significant enhancement to the paper's experimental rigor and provide the best possible comparison within resource constraints.
>
> Please kindly refer to the general feedback for a full disection fo these new results and addition information on the visual instruction tuning baselines.
>
> **Q1:**
> _For Table 3: Why TS training could improve the BBH and GPQA results while harming the MMLU-Pro results? Is it just performance fluctuation or is there an inner difference between these benchmarks?_
>
> **Response:**
> The observed variations arise from differences in task alignment objectives. We would expect the model alignment to shift since we are doing full parameter finetuning. These fluctuations, less than 2%, do not indicate a degradation or improvement in language processing performance and are within expected a reasonable range for model finetuning. Note that the Orca model is finetuned with text-only data and its performance differs from that of the baseline LLama model.
>
> ---
>
> **Q3:**
> _Open question: What affects the time-series data pretraining? e.g., context length, the vision ability, the reasoning ability, etc._
>
> **Response:**
> Time-series data pretraining described in this paper is used as a way to initialize the new time-series token embeddings. The alternative is to take the mean of the existing text embeddings to initialize them. We then pretrain the model on purely time-series data as a way to further refine the added time-series embeddings. Since the loss is only computed on the new time-series embeddings, we have found that this has minimal impact on the natural language performance of these models.

---

> > ### Comment · Reviewer_FVKA · 2024-11-22
> > **Thanks for the response!**
> >
> > I appreciate the additional experiments and explanations, and I'd like to raise my overall assessments. You may want to include these responses to the main contents.
> >
> > However, I'm still worried about the overall model setting. From the given literature in the general response, I find `No Time-Series Data` baselines sometimes surpass those with TS data. Besides, current task settings are closer to TS data comprehension, which cannot fully validate the need of building such a multi-modal LLM. Maybe new tasks like TS data forcasting with TS generation abilities would make more contributions (i.e. input: text instruction, output: TS data).
> >
> > Thank you very much for the response again and best wishes.

---

> > > ### Author Response · Authors · 2024-11-25
> > > **Thank you!**
> > >
> > > Thank you very much for your constructive feedback. We have incorporated as much of the feedback from each reviewer as possible. We thank you for increasing your score and recognizing the novelty and the importance of our contributions. We would like to point out that by training multi-modal LLM’s we can increase their effectiveness in time-series reasoning. It is also our hope that this work will help motivate future work that includes multi-modal LLM’s which are capable of performing all time-series analysis-based tasks and importantly, explain and justify their reasoning, particularly for high-stake applications.
> > >
> > > Thank you again for your feedback!

---

### Author Response · Authors · 2024-11-21
**General Rebuttal**

We appreciate the reviewers' feedback and the opportunity to address their concerns. In response to the comments, we have made significant additions and clarifications. We are currently working in integrating these changes into our submission and are aiming to have them integrated into the paper before the rebuttal deadline.

### 1. **Dataset Quality Check**

To address concerns about the dataset quality:

- We have manually conducted an evaluation on 100 random samples from the dataset to measure the quality, correctness, and relevance. Across these samples, we recorded **92** correct samples. The remaining **8** either had multiple correct answers or the correct answer was unclear and open to interpretation. This shows that overall our synthetically generated datsets is of reasonably high quality.
- We also recalculated performance metrics using this subset. For this subset we took the opportunity to **manually re-write** many of the samples to improve the diversity and accuracy for each sample in the datset. We will refer to this data subset as **TS Instruct QA Gold**. These results reinforce the robustness of the dataset and its minimal impact on model performance due to potential noise.

### 2. **Baseline Additions**

Several new baselines have been added to evaluate the models more comprehensively:

- **Llama-3.2-11B Vision-Instruct**
- **Phi-3.5 Vision-Instruct**
- **Phi-3-medium-4k-instruct**
- **gemma-2-2b-it**
- **gemma-2-9b-it**
- **Ministral-8B-Instruct-2410**
- **GPT-4o-mini**
- **GPT-4o**

These baselines enable a more detailed comparison between models and demonstrate the advantages of using numerical data pretraining for time-series understanding. We also included a direct performance comparison against GPT-4o and GPT-4o-mini, which was heavily involved in dataset creation.

The full results and subsequent analysis is provided below.

---

> ### Author Response · Authors · 2024-11-21
> **General Rebuttal Continued**
>
> ### 3. **QA Gold Task and Vision Model Performance**
>
> We included several new baselines in our evaluation to provide a more comprehensive comparison and better context for the performance of our models. Notably, we added two vision-based models, **Llama-3.2-11B Vision-Instruct** and **Phi-3.5 Vision-Instruct**, as well as strong text-based baselines such as **GPT-4o** and **GPT-4o-mini**. These baselines help highlight the trade-offs between vision-based and text-based approaches for time-series reasoning tasks.
>
> ### Table: Model Performance Metrics
>
> In the table below we show the results of the added base-lines on the new TS Instruct QA Gold dataset. This evaluation shows not only the effects of model size but also of different architectures. It also highlights the importance of work such as ours that focus on building models for time-series reasoning analysis. Despite the size of our model it outperforms model's such as Phi-3-medium-4k-instruct which are both larger and show stronger reasoning capailities on NLP benchmarks. These results also show the importance of scale since scaling up model performance (for example gemma-2b to gemma-9b) generally increases performance on both the MMLU_PRO and time-series benchmarks. We also evaluated powerful "closed-source" models such as GPT-4o and GPT-4o-mini. These results should be interpreted with caution as the GPT-4o-mini (which likely shares training data with GPT-4o) was used in the origional dataset generation process.
>
> Open Source Models
> | Model | Correct Avg % | Correct Std % | Wrong Avg % | Wrong Std % | Null Avg % | Null Std % |
> | -------------------------- | ------------- | ------------- | ----------- | ----------- | ---------- | ---------- |
> | gemma-2-2b-it | 35.80% | 6.53% | 63.80% | 6.22% | 0.20% | 0.45% |
> | gemma-2-9b-it | 43.40% | 8.56% | 54.40% | 8.53% | 2.20% | 1.64% |
> | llama3_1-8b_instruct | 47.40% | 5.41% | 50.40% | 6.80% | 2.20% | 1.64% |
> | Ministral-8B-Instruct-2410 | 49.60% | 5.13% | 50.40% | 5.13% | 0.00% | 0.00% |
> | Phi-3-medium-4k-instruct | 53.60% | 3.21% | 45.20% | 3.77% | 0.80% | 1.79% |
> | PreOrcaTS(ours) | 54.40% | 4.39% | 45.60% | 4.39% | 0.00% | 0.00% |
> | GPT-4o-mini | 66.40% | 3.36% | 33.60% | 3.36% | 0.00% | 0.00% |
> | GPT-4o | 68.20% | 4.39% | 31.80% | 4.39% | 0.00% | 0.00% |
>
> In additional to providing training and testing data, a key contribution of our work was providing a flexible model training process through discrete tokenization which has minimal impact on nlp performance and easily integrates time-series reasoning. The table below shows the importance of training models for time-series analysis. The PreOrcaTS(ours) model performs proportionally better than the other baselines when it comes to the time-series reasoning compared to MMLU_PRO performance. The table is sorted by TS Reasoning performance.
>
> In our updated paper version this will be converted to scatter plot to more clearly visualize these results.
>
> | Model           | TS Performance (Correct Avg %) | MMLU_PRO Score | Category      |
> | :-------------- | -----------------------------: | -------------: | :------------ |
> | gemma-2-2b-it   |                           35.8 |         0.2719 | Open-Source   |
> | gemma-2-9b-it   |                           43.4 |         0.4125 | Open-Source   |
> | llama3_1-8b     |                           47.4 |         0.3772 | Open-Source   |
> | Ministral-8B    |                           49.6 |         0.3483 | Open-Source   |
> | Phi-3-medium-4k |                           53.6 |          0.474 | Open-Source   |
> | PreOrcaTS       |                           54.4 |          0.356 | Multi-modal   |
> | GPT-4o-mini     |                           66.4 |          63.09 | Closed-Source |
> | GPT-4o          |                           68.2 |           76.6 | Closed-Source |

---

> > ### Author Response · Authors · 2024-11-21
> > **General Rebuttal Continued 2**
> >
> > ### Vision Models
> >
> > | Model                         | Correct Avg % | Correct Std % | Wrong Avg % | Wrong Std % | Null Avg % | Null Std % |
> > | ----------------------------- | ------------- | ------------- | ----------- | ----------- | ---------- | ---------- |
> > | Llama-3.2-11B-Vision-Instruct | 67.40%        | 3.36%         | 27.80%      | 4.66%       | 4.80%      | 5.97%      |
> > | Phi-3.5-vision-instruct       | 67.40%        | 3.96%         | 32.80%      | 3.96%       | 0.00%      | 0.00%      |
> >
> > A key contribution of our work is the creation of a multi-modal instruction-tuning and evaluation dataset for time-series reasoning, where the time-series data is derived from **real-world datasets**. This use of real-world time-series ensures that the generated instructions and tasks are grounded in practical, meaningful data. To achieve this, we rely on vision-based models to link the time-series data to corresponding textual instructions. However, this reliance on vision models comes with the trade-off that their performance on our benchmark is likely **inflated**, as they are naturally aligned with the dataset generation process. Another important factor is that the multi-modal adapters of these models benefit from the large volumes of multi-modal training data available which means that the visual adapters are likely more extensively trained in comparison to our multi-modal time-series models.
> >
> > Recent work published after the submission of our paper (Merrill et al., 2024) provides additional context. It evaluates methods for time-series reasoning on a fully **synthetic dataset**, demonstrate that vision-based models do not perform significantly better than text-based approaches and in many cases are objectively worse. Their results show that on synthetic datasets, vision-based models struggle with time-series based reasoning. This suggests that vision-based models may not generalize as effectively to other time-series reasoning datasets, particularly those that are synthetically generated.
> >
> > In summary, while our approach benefits from using **real-world time-series data**—providing practical grounding and making the dataset relevant to real applications—it comes at the cost of being less suitable for the evaluation of vision-based models for time-series analysis. The alternative approach of using fully synthetic datasets, as in recent work, avoids reliance on visual models but sacrifices the real-world grounding of time-series data. Our methodology highlights this important trade-off.
> >
> > ```bibtex
> > @inproceedings{merrill-etal-2024-language,
> >     title = "Language Models Still Struggle to Zero-shot Reason about Time Series",
> >     author = "Merrill, Mike A  and
> >       Tan, Mingtian  and
> >       Gupta, Vinayak  and
> >       Hartvigsen, Thomas  and
> >       Althoff, Tim",
> >     editor = "Al-Onaizan, Yaser  and
> >       Bansal, Mohit  and
> >       Chen, Yun-Nung",
> >     booktitle = "Findings of the Association for Computational Linguistics: EMNLP 2024",
> >     month = nov,
> >     year = "2024",
> >     address = "Miami, Florida, USA",
> >     publisher = "Association for Computational Linguistics",
> >     url = "https://aclanthology.org/2024.findings-emnlp.201",
> >     pages = "3512--3533",
> > }
> > ```

---

### Meta-Review · Area_Chair_8KKR · 2024-12-16

**Metareview:**

This paper presents Chat-TS, a novel approach to integrating time-series data with large language models through a discrete tokenization method and synthetic instruction datasets. The reviewers acknowledge some notable strengths, including the innovative approach to transforming continuous time-series data into discrete tokens, the creation of comprehensive synthetic datasets for time-series reasoning, and the maintenance of performance on text-based tasks (Reviewer FVKA, dxnt). The authors propose three key datasets: a time-series instruction tuning dataset, a time-series QA benchmark, and a quantitative evaluation benchmark (Reviewers xWsL, dxnt).

However, significant weaknesses substantially undermine the paper's contributions. Reviewers consistently raised concerns about the methodological soundness, including the lack of rigorous dataset quality control and potential hallucinations introduced by GPT-4o during dataset generation (Reviewer FVKA). The generalizability of the synthetic datasets is questionable, with limited exploration of performance across diverse time-series types and sequence lengths (Reviewer dxnt). Additionally, the experimental results are mixed, with inconsistent performance across different benchmarks, and the paper lacks comprehensive baseline comparisons and a direct performance comparison with GPT-4o (Reviewers xWsL, dxnt). The presentation was also critiqued for formatting issues and clarity of the dataset creation process. Consequently, the reviewers unanimously recommend rejection, with ratings ranging from "strong reject" to "marginally above the acceptance threshold."

Owing to the violation of the formatting instructions, the paper should be rejected.

**Additional Comments On Reviewer Discussion:**

Reviewers are still questioning the novelty of the method and the experimental results despite the rebuttal.

---

### Decision · Program_Chairs · 2025-01-22

Reject